# Discriminative Visual Process Rewards for Scaling Thinking at Test-Time with Images

Bo-Wen Yin [1 2]  Qize Yang [2]  Boyuan Sun [1]  Xihan Wei [2]  Qibin Hou [1 3]

## Abstract

The "thinking with images" paradigm encourages multimodal large language models to generate intermediate visual steps—such as cropping, annotation, spatial localization, and sketches—to enhance high-resolution perception and complex reasoning. However, existing multimodal Process Reward Models (PRMs) evaluate only textual reasoning and cannot judge the correctness of these visual steps, creating a key gap when visual reasoning is essential for solving tasks. We propose Discriminative Visual Process Reward Model (DiscPRM), a multimodal PRM that jointly evaluates textual and visual intermediate steps by modeling visual reasoning trajectories, image operations, and text-image consistency. To support this, we build VTReward-100K, a dataset of step-by-step visual reasoning sequences with supervision. Experiments show that using DiscPRM for Best-of-N process supervision substantially improves multimodal reasoning performance on tasks requiring visual intermediate steps, achieving over 5% gains across benchmarks. We further introduce VABench, the first benchmark for evaluating PRMs on visual reasoning error detection. We hope this work can provide foundational support for advancing the emerging direction of visual–textual process reward. Project page: https://DiscPRM.github.io.

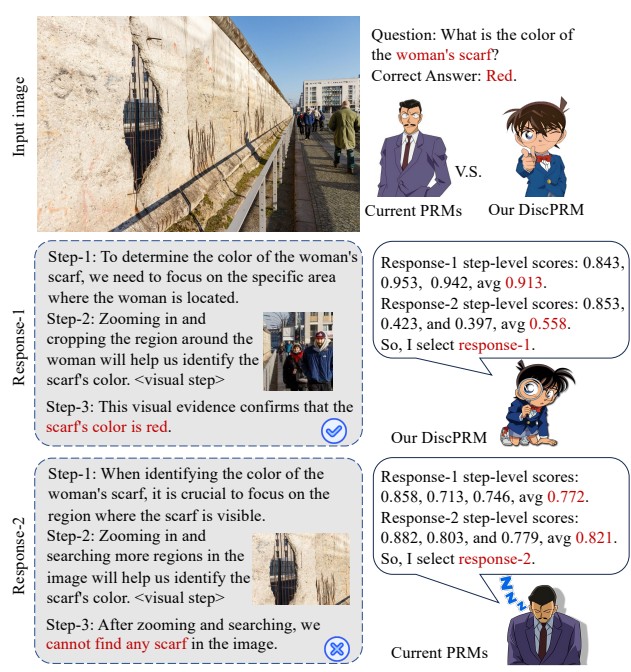

*Figure 1.* Current process reward models (PRMs) v.s. DiscPRM. In thinking-with-images trajectories, existing PRMs often fail to assess key visual reasoning steps; misjudging these steps can cascade to later reasoning and yield unreliable response-level scores. In contrast, DiscPRM explicitly evaluates visual steps, helpful for selecting correct trajectories.

## 1. Introduction

Recent progress in multimodal large language models (MLLMs) (Shen et al., 2025; Xiaomi, 2025; Team, 2025; Chen et al., 2023; Fu et al., 2025) has highlighted the grow-ing importance of *thinking with images* (OpenAI, 2025), a paradigm where models interleave textual reasoning with explicit visual operations—such as cropping relevant regions, adding annotations, performing spatial localization, or sketching auxiliary cues—as part of the reasoning process. These intermediate visual steps (Zheng et al., 2025b; Zhang et al., 2025a) enable models to externalize visual thinking, reduce hallucinations, and produce clearer reasoning trajectories, making visual intermediate reasoning an increasingly central component in modern MLLM design. In parallel, test-time scaling (TTS) (Dong et al., 2024; Snell et al., 2024; Zhang et al., 2025c) improves MLLMs by sampling multiple candidate solutions and selecting the best one, where Process Reward Models (PRMs) (Wang et al., 2024c; Luo et al., 2024) serve as the critic for Best-of-N selection via step-level evaluation.

[1]VCIP, CS, Nankai University [2]Tongyi Lab, Alibaba Group [3]NKIARI, Shenzhen Futian. Correspondence to: Qibin Hou <houqb@nankai.edu.cn>.

*Proceedings of the 43$^{rd}$ International Conference on Machine Learning*, Seoul, South Korea. PMLR 306, 2026. Copyright 2026 by the author(s).

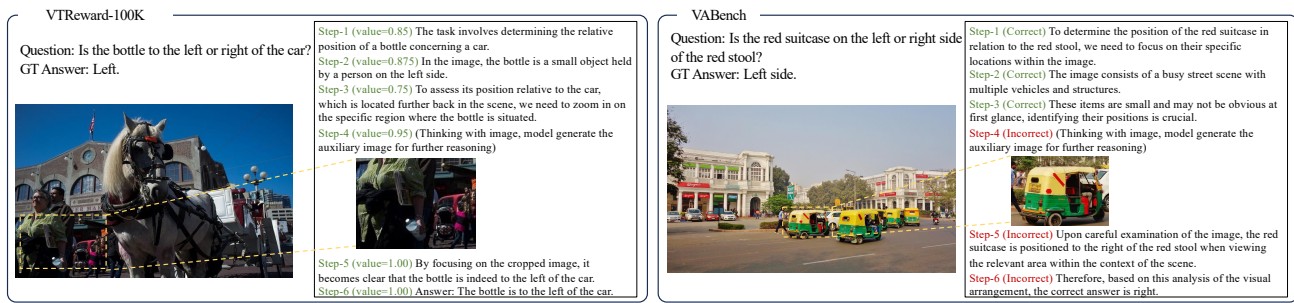

Figure 2. Data examples in VTReward-100K and VABench. Monte carlo tree value estimation method is used to generate the process supervision data and construct the VTReward-100K. Specifically, we estimate the expected accuracy $mc_i$ of the given step $s <= i$ based on Monte Carlo sampling and consider the step correct if $mc_i > 0$. collect questions from diverse existing multimodal reasoning benchmarks (Wang et al., 2025a; Zhang et al., 2024a; Fu et al., 2024) and generate the solutions using leading MLLMs. Based on these questions and solutions, we annotate the correctness of each step in the solutions.

However, extending PRMs to the thinking-with-images setting remains challenging: First, existing PRMs (Luo et al., 2025; Wang et al., 2025b; Tu et al., 2025; Dong et al., 2024) are largely text-centric and lack the capability to assess the correctness of intermediate visual operations, causing them to mis-rank trajectories that rely on visual steps (See Fig. 1); Moreover, there is still no standardized benchmark tailored to evaluating such *visual-aware* critics, and direct BoN-style evaluation is both expensive-requiring many sampled multi-modal trajectories—and confounded by the policy model, making fair comparisons across critics difficult when visual intermediate steps are involved.

To address these challenges, we propose DiscPRM (Discriminative Visual Process Reward Model), a multimodal Process Reward Model that serves as a *visual-aware critic* for *thinking-with-images* trajectories. DiscPRM models visual reasoning trajectories, image manipulations, and text-image consistency to jointly evaluate textual reasoning steps and intermediate visual operations.

To enable training such a critic, we construct VTReward-100K, a dataset of approximately 100K multimodal process-supervision samples with step-level reward annotations. Each sample contains an input image, a question, and a step-by-step solution. In half of the VTReward-100K, the solutions include intermediate visual steps (e.g., crops, annotations, spatial cues, sketches), for which we provide image-grounded supervision. We present an example of VTReward-100K in Fig. 2. The underlying prompts are collected from diverse open datasets, including LLaVA-OV-Image (Li et al., 2024) and MM-RLHF (Zhang et al., 2025b), etc., covering a broad range of multimodal reasoning scenarios. To obtain step-level rewards at scale, we employ an automatic Monte Carlo tree pipeline (Wang et al., 2024c).

During training DiscPRM, each instance is formulated as a multi-turn chat: the first turn contains the image, the question, and the initial reasoning step, while each subsequent turn presents a new visual or textual step. At every

turn, the model is trained to predict the corresponding reward value, allowing it to act as an effective critic in Best-of-N evaluation. After training, DiscPRM can be directly plugged into Best-of-N test-time scaling to select higher-quality reasoning trajectories, leading to consistent improvements when applied to representative thinking-with-images models (Zhang et al., 2025a; Zheng et al., 2025b) on a suite of multimodal benchmarks (Zhang et al., 2024b; Wang et al., 2025a; Wu & Xie, 2024; Wang et al., 2024b; Lu et al., 2024; Fu et al., 2024), and outperforming existing multimodal PRMs (Wang et al., 2025b; Tu et al., 2025) especially on problems where intermediate visual operations are essential.

To support a more faithful evaluation of visual-aware critics beyond expensive and policy-dependent BoN testing, we introduce VABench, the first benchmark targeting step-wise error detection in multimodal reasoning processes with intermediate visual steps. VABench contains 1,000 challenging samples with binary correctness labels for every step (covering both visual and textual reasoning), and is built via a hybrid annotation pipeline that combines Monte Carlo tree exploration for candidate error discovery, calibration by multiple commercial MLLMs , and final human verification. Unlike prior benchmarks (Lightman et al., 2023; Zheng et al., 2025a) that focus only on the first mistake, VABench requires detecting all erroneous steps in a trajectory, providing a more comprehensive and reflection-compatible assessment of PRMs' capability to evaluate thinking-with-images reasoning. Evaluation results reveal that existing open-source multi-modal PRMs struggle to reliably judge step-wise correctness when visual reasoning is involved, underscoring the necessity of our visual-aware PRM, DiscPRM.

In summary, our main contributions are as follows.

- We develop DiscPRM, a visual-aware PRM tailored for thinking with images, which substantially improves Best-of-N selection for vision-intensive reasoning tasks, boosting the performance of state-of-the-art thinking with images models and outperforming existing multimodal

PRMs across multiple benchmarks.

- We present VTReward-100K, a multimodal process-supervision dataset with step-level rewards for both textual and visual intermediate reasoning, to support the training of visual-aware PRM.
- We construct VABench, the first benchmark for evaluating PRMs on detecting errors in multimodal reasoning trajectories containing intermediate visual steps.

## 2. Related Work

**Thinking with Images.** Thinking with images (OpenAI, 2025; Team, 2025; Chen et al., 2023; Fu et al., 2025) is an emerging paradigm in MLLMs, where models interleave language reasoning with intermediate visual operations (*e.g.,* cropping, annotation, localization, sketches) to support problem solving. Recent systems (Zheng et al., 2025b; Zhang et al., 2025a; Zhao et al., 2025) explore different realizations, including generating auxiliary images, selecting informative regions via bounding boxes/crops, and using tool/code-driven pipelines to perform richer image manipulations. While these methods demonstrate the benefits of explicit visual reasoning trajectories, further scaling their reasoning ability becomes more and more vital. However, current test-time scaling methods (Dong et al., 2024; Luo et al., 2024; Snell et al., 2024) are still largely text-centric and provide limited supervision or verification for the correctness of intermediate visual steps.

**Process Reward Models.** Process reward models (PRMs) (Wang et al., 2025b; Tu et al., 2025; Dong et al., 2024) have become a central component in reinforcement learning (RL) (Ahmadian et al., 2024; Schulman et al., 2017) optimization and test-time scaling (TTS) (Wang et al., 2024c). Unlike Outcome Reward Models (ORMs) (McAleese et al., 2024; Wang et al., 2024a), which provide a single holistic score for the final response, PRMs evaluate the quality of each intermediate reasoning step and aggregate these step-level scores into a final reward. This finer-grained supervision not only mitigates "spurious correctness," where an incorrect reasoning path leads to an accidentally correct answer, but also yields more stable and interpretable reasoning behavior. Prior work ranges from human-annotated process supervision (Lightman et al., 2023) to automatic pipelines (Wang et al., 2024c; Chen et al., 2025) using Monte Carlo sampling and LLM-based judgments. Current PRMs (Wang et al., 2025b; Tu et al., 2025; Chen et al., 2025) extends to multimodal inputs but remains focused on scoring textual reasoning steps. In contrast, our work targets thinking-with-images trajectories and trains a visual–textual PRM that can assess both visual operations and textual reasoning within a unified framework.

**Benchmarks for Reward Models.** The evaluation of reward models (RMs) has received increasing attention as RMs become central to RL and test-time scaling pipelines. Early benchmarks (Lambert et al., 2025; Li et al., 2025; Liu et al., 2024) primarily focus on binary preference judgment, comparing a chosen and rejected response pair to assess RM quality. Subsequent work (Zhou et al., 2024) extends these evaluations to pairwise and Best-of-N settings, enabling a more comprehensive assessment. With the rise of PRMs, several benchmarks (Song et al., 2025; Wang et al., 2025b; Zheng et al., 2025a) have further been proposed to evaluate step-wise judgment capabilities, allowing finer inspection of a model's ability to detect reasoning errors within a solution. However, despite these advancements, all existing benchmarks remain fundamentally text-only, which is no longer adequate for modern multimodal LLMs that adopt the thinking-with-images paradigm. To address this limitation, we propose VABench, the first benchmark designed to evaluate PRMs and MLLMs on detecting erroneous visual intermediate reasoning steps, to fill a key methodological gap and establishes a foundation for advancing visual-aware PRMs under the thinking-with-images paradigm.

## 3. Scaling Thinking with Images

### 3.1. DiscPRM

First, we briefly describe the setting of the research problem. For a given problem $P$ with input image $I$, thinking with images models can generate solution $S$. Unlike conventional MLLM solutions that are purely textual, a thinking-with-images solution interleaves language steps with visual steps produced along the trajectory; these intermediate images are treated as part of the context and encoded into embeddings in the same way as the input image. Each solution is composed of visual and textual steps, denoted as $S = (s_1, s_2, ..., s_n)$, where $s_i$ represents the $i$-th step. Accordingly, we assign a step-wise score to each step—both visual and textual—so that the critic can evaluate the entire multimodal reasoning process. To achieve test-time scaling of the thinking with images models, we conduct process evaluation on multiple (N) solutions and select the best one, *i.e.,* Best-of-N evaluation. Specifically, process reward models (PRMs) conduct the process evaluation of the solutions via a multi-turn dialogue manner, and the probability value predicted by token '+' is used as the score of the step. The first step $s_0$ of the solution to this question is included in the first turn and a new step is presented in each subsequent turn. PRM is required to evaluate the score of the given step in each turn as follows:

$$v_i \sim M(v_i \mid I, P, s_{\leq i}), \qquad (1)$$

where $v_i$ denotes the score of $i$-th step.

In general, PRMs can be categorized into two types, *i.e., value-based* and *advantage-based*. A value-based PRM assigns each step a quality score based on its expected ac-

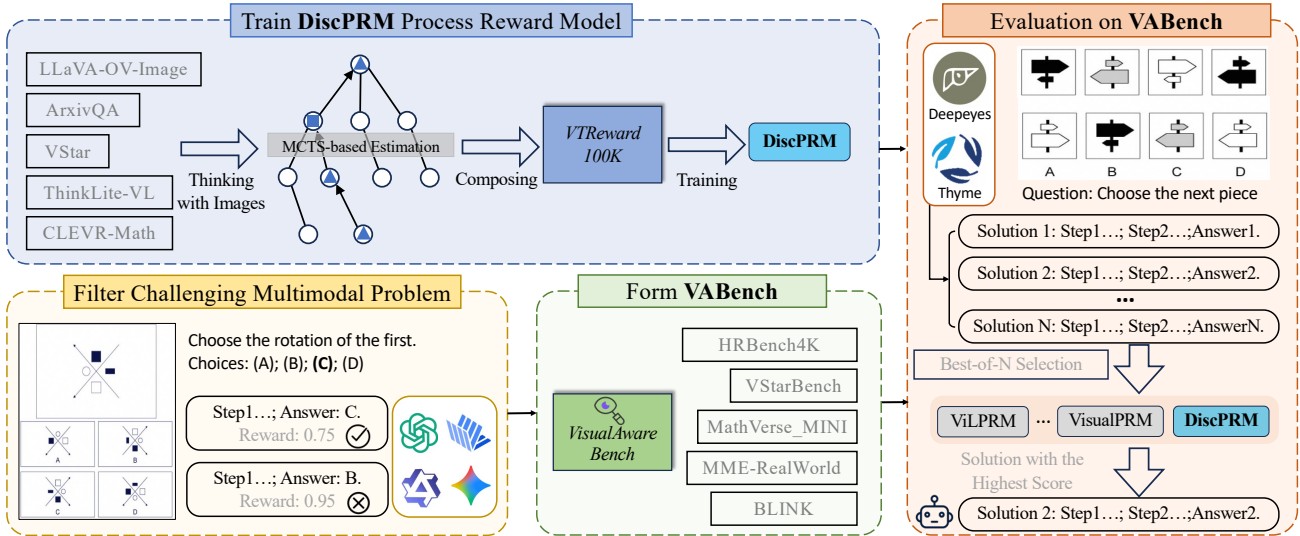

*Figure 3.* We present a suite of collaborative visual-textual process reward modeling for thinking with images. Firstly, we generate thinking with images solutions on a large number of diverse visual language problems and obtain step-level reward annotation via MCTS-based estimation, composing our VTReward-100K. Then, we tune Qwen2.5-VL-7B on the VTReward-100K to acquire DiscPRM. To evaluate the capacity of PRMs on visual-textual solutions, we filter challenging problems, annotate the responses, and construct the VABench. Finally, we evaluate current popular MLLMs, state-of-the-art PRMs and our DiscPRM on VABench.

curacy $mc_i \in [0, 1]$, whereas an advantage-based PRM measures the improvement of expected accuracy over the previous step (*i.e.*, $mc_i - mc_{i-1}$), analogous to the advantage function, and typically discretizes the step quality into $\{+, =, -\}$ indicating a superior, comparable, or inferior situation. Empirically, we find the value-based formulation works better for our DiscPRM, and thus we adopt it by default. Specifically, following Math-Shepherd (Wang et al., 2024c; Dong et al., 2024), instead of regressing the exact $mc_i$, we train the model to predict a discretized correctness label $c_i \in \{+, -\}$ for each step, where the $i$-th step is labeled as positive if $mc_i > 0$ and negative otherwise. We always supervise all steps in the trajectory as well as current state-of-the-art PRMs (Wang et al., 2025b; Tu et al., 2025).

During inference, we compute the response-level score for multiple responses and select the best one, achieving test-time scaling of thinking with images. To be particular, DiscPRM takes weighted sum of the generation probability for the discretized scores as the value of each step. Then, DiscPRM averages each step's value as the response score:

$$R(S) = \frac{1}{n}\sum_{i=1}^{n} 1 \cdot p(c_i = + \mid I, Q, s_{\leq i}) \quad (2)$$
$$+ 0 \cdot p(c_i = - \mid I, Q, s_{\leq i}),$$

where the scores of labels $\{+, -\}$ are $\{1, 0\}$, $p(c_i = + \mid I, Q, s_{\leq i})$ and $p(c_i = + \mid I, Q, s_{\leq i})$ are the generation probability for labels $\{+, -\}$ respectively.

### 3.2. VTReward-100K

We present the data example of our VTReward-100K in Fig. 2. Specifically, each sample contain an image $I$, a problem $P$, a step-by-step solution $S$, and the annotated process correctness value $V = (v_1, v_2, ..., v_n)$, $v_i \in [0, 1]$ for each step, where $v_i$ denotes the expected accuracy of the $i$-th step and is automatically estimated via a Monte Carlo rollout pipeline (Wang et al., 2024c).. To ensure data diversity, we collect the image and question pairs from various tasks from LLaVA-OV-Image (Li et al., 2024) and MM-RLHF (Zhang et al., 2025b), etc. The step-by-step solutions $S$ are sampled using thinking with images models (Zheng et al., 2025b; Zhang et al., 2025a).

The key idea is to estimate the *expected* probability that the solution will finally reach the correct answer when conditioning on the current prefix $S_{\leq i} = (s_1, ..., s_i)$. Concretely, for each prefix $S_{\leq i}$, we sample $K$ continuations by letting a policy model $M$ complete the remaining steps:

$$\tilde{S}_{>i}^{(k)} \sim M(\tilde{S}_{>i} \mid I, Q, S_{\leq i}), \quad k = 1, ..., K, \quad (3)$$

where each completion $\tilde{S}_{>i}^{(k)}$ may also contain mixed visual and textual steps. We then execute/parse the completed trajectory to obtain its final answer and compare it with the ground-truth answer to determine whether the completion is correct. The step value is computed as the Monte Carlo estimate:

$$v_i = \frac{1}{K}\sum_{k=1}^{K} \mathbb{I}\Big[\text{Ans}\Big(S_{\leq i} \oplus \tilde{S}_{>i}^{(k)}\Big) = \text{GT}\Big], \quad (4)$$

| | Dataset Souce | Size | Split | Ori. Size |
|---|---|---|---|---|
| Composition | HRBench4K | 200 | Val | 800 |
| | VStarBench | 200 | Val | 191 |
| | MathVerse | 200 | Vision Only | 788 |
| | MME-Real | 200 | Val | 11,915 |
| | BLINK | 200 | Val | 1,901 |
| | Sum | 1000 | Test | 15,595 |

*Table 1.* Composition overview of VABench. 'Ori. Size' represents the original size of the dataset.

which measures how likely the current step prefix can be extended into a correct full solution. Intuitively, a step receives a high value if it keeps the reasoning trajectory on track, and a low value if it leads to failure even after multiple sampled continuations.

To construct VTReward-100K, we sample 4 solutions for each image-question pair and split each of them into step-by-step format. For each step, we take all the previous steps as prefix, then sample 16 continuations compute $m_i$ according to these continuations. The resulting dataset comprises approximately 100K samples and nearly 1 million steps with process supervision. Each response averages 135.4 words and 9.7 steps, while each step averages 26.2 words. Among these steps, about 15% are incorrect steps.

### 3.3. VABench

To address the lack of standardized evaluation for *visual-aware* process critics, we introduce **VABench**, a benchmark designed specifically for measuring whether PRMs can reliably assess *thinking-with-images* trajectories that interleave textual reasoning with intermediate visual operations, as illustrated in Fig. 2. Each benchmark instance contains image $I$, problem $P$, a step-by-step solution $S$, and a binary correctness label for every step $v_i$ covering both textual and visual steps. Unlike prior benchmarks that focus on text-only solutions, our benchmark explicitly includes intermediate visual steps, enabling a faithful evaluation of PRMs on thinking-with-images trajectories and their ability to judge step-wise correctness in both visual and textual reasoning. As shown in Tab. 1, we build VABench by collecting challenging questions from representative multimodal reasoning benchmarks (*i.e.,* HRBench4K (Wang et al., 2025a), VStar-Bench (Wu & Xie, 2024), MathVerse (Zhang et al., 2024a), MME-RealWorld (Zhang et al., 2024b), and BLINK (Fu et al., 2024).) and sampling diverse step-by-step solutions from Thyme and Deepeyes. To obtain reliable step labels at scale, we employ a hybrid annotation pipeline: (i) we first run a Monte Carlo tree procedure to expand alternative continuations and automatically surface suspicious steps that likely cause downstream failure; (ii) we then calibrate these preliminary labels with multiple commercial MLLMs, *i.e.,* Qwen-VL3-Max, Gemini 3 Pro, and GPT-5.2, to refine the step-wise correctness estimates; and (iii) finally, human

reviewers manually inspect and validate the synthesized annotations to ensure high-quality binary step correctness. The resulting benchmark contains 1000 solutions with 9885 total steps, where half are visual-textual solutions and half are textual solutions, and we report step-level error detection performance on VABench in Section 4. We use macro F1 scores to compare model performance, *i.e.,* first compute the F1 scores separately for correct and incorrect steps and then take their average to obtain the overall score.

## 4. Experiments

### 4.1. Test-time Scaling Performance

**Experiment Settings.** DiscPRM is built upon Qwen2.5-VL-7B (Bai et al., 2025) and we use the pre-trained weights of QwenVL-2.5-7B, following the common practice of PRMs. We formalize the model input and output as: given the input problem and the model reasoning response, the score head transforms the logits feature of the last token of each step into a scalar. This scalar value serves as the predicted reward score for each step and the average score of all the steps is seen as the score of this response. We use the vanilla Mean Square Error (MSE) loss between the ground truth reward and the predicted one to update the DiscPRM. Then, we use DiscPRM as the critic model for Best-of-N evaluation to achieve test-time scaling of thinking with images. We set $N$ to 8 as in (Wang et al., 2025b; Tu et al., 2025; Chen et al., 2025). We sample $N$ distinct thinking with images solutions from the policy models with a temperature of 0.7. The PRM assign each solution with a score and the one with the highest score is then selected to determine the correctness.

**Benchmarks.** For the evaluation of the reasoning abilities of MLLMs, we adopt a range of benchmarks covering Perception, Reasoning, and General tasks. For the perception tasks, the benchmarks include HRBench4K/8K (Wang et al., 2025a), MME-Real World/-CN/-Lite (Zhang et al., 2024b), VStarBench (Wu & Xie, 2024), and RealWorld QA (xAI, 2024). For the reasoning tasks, we use MathVision (Wang et al., 2024b), MathVista (Lu et al., 2024), MathVerse (Zhang et al., 2024a), LogicVista (Xiao et al., 2024), WeMath (Qiao et al., 2024), and VisuLogic (Xu et al., 2025). Among them, we report Vision Only split for MathVerse and MINI split for MathVision and MathVista. MMStar (Chen et al., 2024a), MMVet (Yu et al., 2024), OCR Bench (Liu et al., 2023), Chart QA (Masry et al., 2022), and BLINK (Fu et al., 2024) are used for the evaluation on general tasks.

**Comparison with Other MLLMs.** As shown in Tab. 2, our DiscPRM consistently enhances the abilites of thinking with images across all the benchmarks. Specifically, equipped with DiscPRM, the performance of Deepeyes and Thyme improves by 6.0, 6.6, 5.4, and 5.5, 6.2, 6.0 on reasoning,

| | Model | MathVista | MathVision | MathVerse-VO | WeMath | LogicVista | VisuLogic | Overall |
|---|---|---|---|---|---|---|---|---|
| **Reasoning Benchmarks** | GPT-4o | 63.4 | 36.5 | 35.3 | 44.2 | 53.2 | 26.7 | 43.2 |
| | InternVL3-8B | 70.4 | 26.3 | 29.2 | 31.7 | 45.6 | 24.9 | 38.0 |
| | Qwen2.5-VL-7B | 68.2 | 27.0 | 35.2 | 34.3 | 39.8 | 20.0 | 37.4 |
| | Qwen2.5-VL-32B | 72.2 | 35.2 | 40.0 | 54.4 | 47.1 | 25.8 | 45.8 |
| | Deepeyes | 70.1 | 26.6 | 47.3 | 38.9 | 47.7 | 22.5 | 42.2 |
| | ViLPRM | 72.6 | 32.5 | 48.8 | 41.1 | 50.7 | 23.7 | 44.9 |
| | VisualPRM | 72.9 | 31.9 | 49.1 | 41.9 | 51.8 | 24.5 | 45.4 |
| | DiscPRM | 74.1 | 36.4 | 52.1 | 43.7 | 54.3 | 25.8 | 47.7 |
| | Thyme-7B | 70.0 | 27.6 | 39.1 | 39.3 | 49.0 | 23.4 | 41.4 |
| | ViLPRM | 72.3 | 30.0 | 39.9 | 41.8 | 53.5 | 23.9 | 43.6 |
| | VisualPRM | 71.7 | 29.7 | 41.6 | 42.2 | 52.6 | 24.9 | 43.8 |
| | DiscPRM | 75.5 | 37.1 | 46.0 | 44.6 | 54.9 | 26.3 | 47.4 |

| | Model | HRbench-4K | HRbench-8K | MME-Real | -CN | -Lite | V* | RealWorld QA | Overall |
|---|---|---|---|---|---|---|---|---|---|
| **Perception Benchmarks** | GPT-4o | 65.0 | 59.6 | 62.8 | 59.7 | 52.0 | 67.5 | 75.5 | 63.2 |
| | InternVL3-8B | 70.0 | 69.3 | 61.3 | 60.5 | 48.6 | 70.2 | 70.0 | 64.3 |
| | Qwen2.5-VL-7B | 68.8 | 65.3 | 58.3 | 60.8 | 44.1 | 76.4 | 68.2 | 63.1 |
| | Qwen2.5-VL-32B | 73.4 | 70.4 | 61.0 | 60.5 | 46.2 | 81.2 | 70.2 | 66.1 |
| | Deepeyes | 75.1 | 72.6 | 63.3 | 62.1 | 51.9 | 90.1 | 69.9 | 69.3 |
| | ViLPRM | 76.5 | 74.1 | 65.5 | 65.1 | 54.3 | 90.5 | 71.1 | 71.0 |
| | VisualPRM | 77.1 | 74.8 | 66.5 | 67.0 | 52.8 | 90.7 | 71.3 | 71.5 |
| | DiscPRM | 80.7 | 78.4 | 72.8 | 67.9 | 58.4 | 95.0 | 75.3 | 75.5 |
| | Thyme-7B | 77.0 | 72.0 | 64.8 | 64.6 | 55.2 | 82.2 | 70.2 | 69.4 |
| | ViLPRM | 79.3 | 76.1 | 65.2 | 67.8 | 59.2 | 83.3 | 71.5 | 71.8 |
| | VisualPRM | 78.9 | 74.8 | 68.5 | 67.1 | 58.6 | 84.0 | 71.9 | 72.0 |
| | DiscPRM | 82.1 | 78.8 | 73.2 | 70.2 | 61.7 | 89.5 | 76.4 | 76.0 |

| | Model | MMStar | MMVet | OCR Bench | Chart QA | BLINK | Overall |
|---|---|---|---|---|---|---|
| **General Benchmarks** | GPT-4o (OpenAI, 2024) | 65.7 | 58.3 | 80.9 | 85.7 | 63.3 | 70.8 |
| | InternVL3-8B (Zhu et al., 2025) | 68.5 | 55.1 | 88.1 | 85.9 | 55.5 | 70.6 |
| | Qwen2.5-VL-7B (Bai et al., 2025) | 64.7 | 52.9 | 88.4 | 83.7 | 56.4 | 69.2 |
| | Qwen2.5-VL-32B (Bai et al., 2025) | 69.1 | 48.4 | 85.5 | 81.1 | 63.6 | 69.5 |
| | Deepeyes (Zheng et al., 2025b) | 64.3 | 57.8 | 83.9 | 85.2 | 56.3 | 69.5 |
| | ViLPRM (Tu et al., 2025) | 66.7 | 58.5 | 85.7 | 86.9 | 58.0 | 71.2 |
| | VisualPRM (Wang et al., 2025b) | 67.3 | 59.2 | 85.1 | 87.0 | 58.7 | 71.5 |
| | DiscPRM | 71.7 | 60.9 | 88.4 | 88.7 | 67.7 | 75.5 |
| | Thyme-7B (Zhang et al., 2025a) | 65.9 | 58.3 | 86.3 | 86.1 | 56.1 | 70.5 |
| | ViLPRM (Tu et al., 2025) | 67.7 | 59.2 | 88.0 | 87.9 | 62.1 | 73.0 |
| | VisualPRM (Wang et al., 2025b) | 68.2 | 60.3 | 87.5 | 87.2 | 59.6 | 72.6 |
| | DiscPRM | 73.3 | 60.1 | 88.2 | 89.5 | 68.3 | 75.9 |

*Table 2.* **Performance on Perception, Reasoning, and General Tasks** from top to bottom. The overall score is the average score of the above benchmarks. We compare DiscPRM against ViLPRM and VisualPRM, using Deepeyes and Thyme as the policy models. By using DiscPRM as the critic model, existing open-source MLLMs achieve significant improvements in reasoning ability under the Best-of-8 evaluation strategy.

perception, and general tasks. These improvements significantly surpasses that of VisualPRM, *i.e.,* 2.4, 2.6, 2.1 and 3.2, 2.2, 2.0. In Fig. 4, we present an example of using our DiscPRM and VisualPRM for selecting the best thinking with images solutions. In the example, VisualPRM is unable to distinguish the advantages and disadvantages of visual processes in different reasoning intermediate processes and assigns these two steps with similar scores. However, for thinking with images, the intermediate visual process usually plays a decisive role, *i.e.,* appropriate one leads to the correct answer and vice versa. This situation illustrates that current PRMs like VisualPRM are not capable of distinguishing the accuracy of thinking with images responses and may select an incorrect solution. We also present additional examples in the Appendix, verifying that the proposed DiscPRM performs well on diverse tasks.

### 4.2. Performance on VABench

Because PRMs and conventional MLLMs have different ways of handling response steps, there will also be differences in the evaluation process, *i.e.,* PRMs will output the likelihood of the step being correct after each step, while MLLMs can only determine whether the step is correct or incorrect. For the evaluation of PRMs including our DiscPRM, each step is seen as correct if the probability of outputting '+' exceeds a certain threshold of 0.5. For the evaluation of MLLMs, the model is prompted to analyze each step and determine its correctness, classifying it as either correct or incorrect.

As shown in Tab. 3, most existing MLLMs struggle to accurately assess the correctness of reasoning process in thinking with images. We take random guessing as baseline and its

| Model | HRBench4K | VStarBench | MathVerse-VO | MME-RealWorld | BLINK | Overall |
|---|---|---|---|---|---|---|
| Random Guessing | 50.0 | 50.0 | 50.0 | 50.0 | 50.0 | 50.0 |
| *Proprietary Models* | | | | | | |
| GPT-4o-Mini (OpenAI, 2024) | 58.2 | 54.4 | 59.3 | 58.0 | 53.7 | 56.7 |
| GPT-4o (OpenAI, 2024) | 64.8 | 57.6 | 58.9 | 62.2 | 55.7 | 59.8 |
| Gemini-2.0-Flash (Deepmind, 2024) | 59.2 | 63.3 | 64.1 | 61.7 | 57.4 | 61.1 |
| Gemini-2.5-Pro (Comanici et al., 2025) | 63.3 | 65.4 | 62.8 | 68.1 | 67.1 | 65.3 |
| Gemini-3-Pro (Deepmind, 2025) | 66.2 | 69.4 | 63.5 | 65.7 | 67.5 | 66.5 |
| *Open-source Models* | | | | | | |
| LLaVA-OV-7B (Li et al., 2024) | 48.8 | 49.6 | 52.4 | 57.3 | 50.7 | 51.8 |
| LLaVA-OV-72B (Li et al., 2024) | 51.8 | 50.3 | 54.3 | 58.8 | 51.2 | 53.3 |
| Qwen2.5-VL-7B (Bai et al., 2025) | 52.9 | 51.9 | 54.3 | 53.8 | 55.1 | 53.6 |
| Qwen2.5-VL-32B (Bai et al., 2025) | 55.2 | 58.2 | 55.9 | 54.9 | 57.3 | 56.3 |
| Qwen2.5-VL-72B (Bai et al., 2025) | 57.8 | 61.0 | 54.7 | 60.7 | 57.5 | 58.3 |
| Qwen3-VL-8B (Yang et al., 2025) | 56.7 | 59.8 | 54.9 | 62.1 | 60.3 | 58.8 |
| InternVL2.5-8B (Chen et al., 2024b) | 44.3 | 49.7 | 52.5 | 56.8 | 49.9 | 50.6 |
| InternVL2.5-38B (Chen et al., 2024b) | 50.1 | 48.8 | 54.1 | 55.6 | 50.9 | 51.9 |
| InternVL2.5-78B (Chen et al., 2024b) | 51.9 | 49.7 | 55.9 | 57.7 | 51.2 | 53.3 |
| InternVL3-8B (Zhu et al., 2025) | 53.4 | 55.1 | 58.8 | 60.2 | 56.3 | 56.8 |
| *Process Reward Models* | | | | | | |
| ViLPRM (Tu et al., 2025) | 59.7 | 63.9 | 62.0 | 57.3 | 53.5 | 59.3 |
| VisualPRM (Wang et al., 2025b) | 59.2 | 62.3 | 61.2 | 59.7 | 57.1 | 59.9 |
| DiscPRM (ours) | 65.1 | 64.8 | 66.5 | 64.3 | 68.9 | 65.9 |

*Table 3.* **Results on VABench.** We report the macro F1 of the correct and incorrect steps. The overall score is the micro average of the score from different data sources. Our DiscPRM exhibits state-of-the-art performance among open-source models.

| Critic Model | MathVista | HRBench4K | MMStar | VABench |
|---|---|---|---|---|
| Pass@1 | 70.0 | 77.0 | 65.9 | - |
| Random | 68.0 | 76.6 | 64.7 | 50.0 |
| Qwen2.5-VL-7B | 70.3 | 77.8 | 65.4 | 53.6 |
| Qwen2.5-VL-32B | 70.5 | 77.9 | 66.0 | 56.3 |
| Advantage-based formulation | | | | |
| +Min | 71.4 | 79.1 | 70.1 | 59.6 |
| +Max | 71.1 | 78.5 | 70.3 | 59.6 |
| +Average | 72.4 | 79.9 | 71.1 | 59.6 |
| Value-based formulation (w. early stop) | | | | |
| +Min | 72.0 | 80.3 | 70.2 | 63.4 |
| +Max | 71.4 | 79.2 | 69.8 | 63.4 |
| +Average | 73.5 | 80.5 | 71.5 | 63.4 |
| Value-based formulation (w/o early stop) | | | | |
| +Min | 73.5 | 79.7 | 71.7 | 65.9 |
| +Max | 73.1 | 79.5 | 72.0 | 65.9 |
| +Average | 75.5 | 82.1 | 73.3 | 65.9 |

*Table 4.* **Comparison of different critic models and score aggregation methods.** Note that score aggregation methods do not affect performance on VABench, which focuses on step correctness judgement. We find that supervising all steps (w/o early stop) during training perform slightly better than supervising only up to the first incorrect step (w. early stop).

overall F1 score is 50.0. Our method significantly outperforms all open source models, including other PRMs. We manually check the judgments of other PRMs and observe that these models tend to provide positive analysis and label most visual steps as correct. Thus, the steps after a wrong visual step tend to be wrong, and these PRMs still give high scores to these steps. For example, VisualPRM achieves

an F1 score of 87.1 for positive steps, while its F1 score for negative steps is only 32.7, indicating that VisualPRM rarely identifies incorrect steps. Furthermore, compared to proprietary models, our DiscPRM demonstrates competitive performance, achieving an overall score of 65.9— outperforming GPT-4o and GPT-4o-Mini, and performing on par with Gemini-2.5-Pro and Gemini-3-Pro. Notably, our model, with only 7 billion parameters, is more efficient than these proprietary counterparts.

### 4.3. More Analysis

**Reward Modeling Manners.** We conduct an ablation study on two reward modeling manners, *i.e.,* value-based and advantage-based, and on different ways to aggregate step-level scores into response-level, including mean pooling, max pooling, and min pooling. As summarized in Tab. 4, the value-based formulation consistently performs better than the advantage-based one across both BoN evaluation and VABench. We also compare supervising all steps in a trajectory versus stopping supervision at the first incorrect step; the all-steps strategy yields stronger results. Finally, among aggregation rules, max pooling performs worst others. A likely explanation is that many solutions start with generic, high-confidence steps that receive near-perfect scores, while errors tend to occur later; consequently, taking the maximum over steps is dominated by these early high scores. In contrast, averaging leverages information from multiple

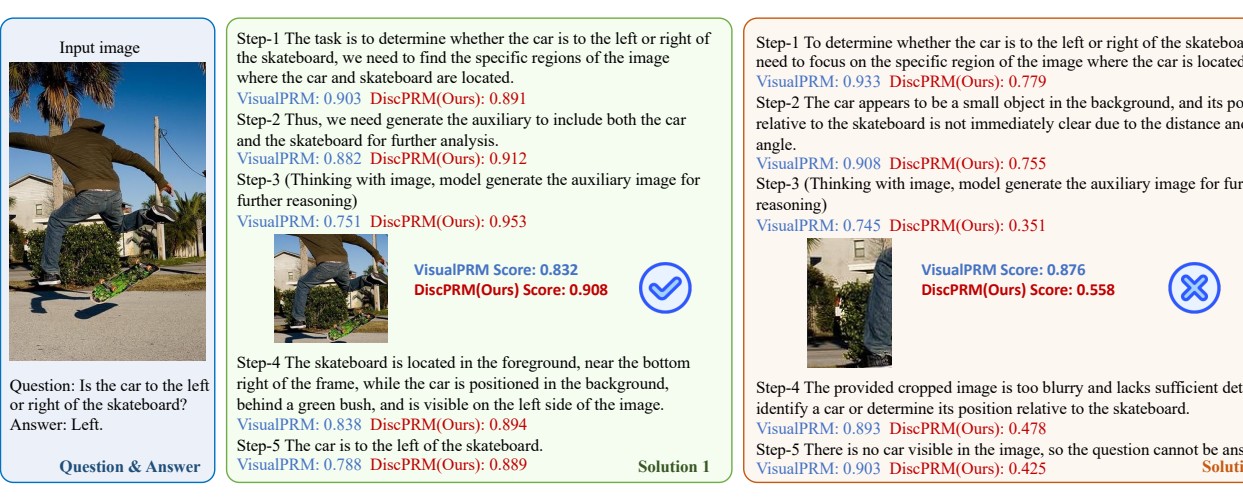

*Figure 4.* An example of process scores provided by VisualPRM (Wang et al., 2025b) and our DiscPRM. We mark different scores with different colors. Current PRMs like VisualPRM cannot effectively judge the correctness of visual steps, creating a key gap when evaluating thinking with images solutions.

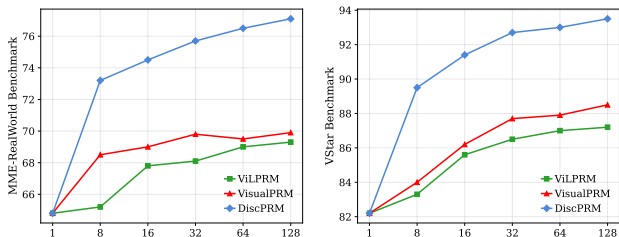

*Figure 5.* Best-of-N results with different sampling numbers. Here we use Thyme as policy model.

*Table 5.* Effect of different sampling temperatures used in constructing VTReward-100K. We report Best-of-8 performance using DiscPRM with Thyme and Deepeyes as policy models. Moderate temperatures achieve the best trade-off between trajectory diversity and reliability.

| Temperature | V* | MathVision | BLINK |
|---|---|---|---|
| DiscPRM (0.4) | 88.8 | 36.5 | 67.8 |
| DiscPRM (0.7) | 89.5 | 37.1 | 68.3 |
| DiscPRM (1.0) | 89.0 | 36.9 | 68.4 |
| DiscPRM (1.3) | 88.3 | 36.3 | 68.0 |

steps and is more robust, effectively acting as an ensemble over step-level estimates.

**Effect of Sampling Number.** To evaluate our method under different selection settings, we increase the number of sampled solution candidates and choose the final answer using VisualPRM, ViLPRM, and our DiscPRM. As shown in Fig. 5, in BoN evaluation, increasing the sampling number of solutions improves the thinking with images ability when using the three models, with DiscPRM yielding the most significant improvements. For example, with Thyme as the policy model, our visual-aware selection via DiscPRM consistently yields higher accuracy than VisualPRM under Best-of-N evaluation on MME-RealWorld, *e.g.,* +4.7 points at N=8. As N increases, the advantage of DiscPRM becomes more pronounced, reaching +7.2 points at N=128. In contrast, while other PRMs can bring gains at small N, their improvements are not monotonic with larger candidate pools (*e.g.,* Best-of-128 is comparable with Best-of-64). We hypothesize increasing the sampling budget in thinking-with-images introduces more diverse visual operation trajectories that can lead to the correct answer, but existing text-centric critics struggle to reliably recognize and reward these critical visual steps. Overall, these results demonstrate

that DiscPRM provides an effective and scalable critic for test-time scaling of thinking with images.

**Effect of Different Temperatures.** We further analyze the robustness of DiscPRM under different sampling strategies used during the construction of VTReward-100K. The sampling temperature controls the diversity and reliability of the generated thinking-with-images trajectories: a lower temperature tends to produce more deterministic and conservative solutions, while a higher temperature increases trajectory diversity but may also introduce noisier or less reliable intermediate visual steps. To study this effect, we construct VTReward-100K variants using different temperatures, including 0.4, 0.7, 1.0, and 1.3, and evaluate the resulting DiscPRM under Best-of-8 selection with both Thyme and Deepeyes as policy models.

As shown in Tab. 5, the performance of DiscPRM remains generally stable across a wide range of temperatures. For Thyme, the results vary only mildly from 88.3 to 89.5 on V*, 36.3 to 37.1 on MathVision, and 67.8 to 68.4 on BLINK. A similar trend can be observed with Deepeyes, where the performance stays within a narrow range across all three

benchmarks. These results indicate that DiscPRM is not overly sensitive to the exact sampling temperature used for data construction, suggesting that the proposed visual-textual process supervision is robust to moderate variations in trajectory sampling.

Among all settings, the moderate temperature of 0.7 achieves the best overall trade-off. Compared with the lower temperature of 0.4, temperature 0.7 provides more diverse reasoning trajectories and intermediate visual operations, which helps DiscPRM learn to distinguish correct and incorrect visual reasoning steps more effectively. In contrast, increasing the temperature to 1.0 or 1.3 does not bring further improvements and sometimes slightly degrades performance, likely because overly stochastic sampling introduces more noisy trajectories and unreliable visual operations. Therefore, we adopt temperature 0.7 as the default setting for constructing VTReward-100K in our experiments.

**Analysis to Other PRMs.** Existing MLLMs can be prompted to act as critic models, but we observe that they struggle to reliably distinguish good vs. bad steps in *thinking-with-images* trajectories, and thus bring only marginal gains under Best-of-N selection, as shown in Tab. 2. Compared with these prompted critics, prior PRMs can improve performance to some extent, yet they remain significantly weaker than our approach, as shown in Tab. 2 and Tab. 3. We attribute this gap to their text-centric nature: while they can often judge the correctness of textual reasoning, they frequently fail to assess the validity of key visual reasoning steps, *e.g.,* whether an intermediate crop/annotation actually captures the necessary evidence, and tend to assign overly positive scores to such steps. Since these visual operations are often decisive—correct visual grounding typically makes the final answer much more likely to be correct, whereas an incorrect visual step usually leads to failure—mis-scoring them causes error propagation: even after an erroneous visual step, existing PRMs may still output high scores and continue producing positive judgments for later steps, making them unreliable for evaluating and ranking thinking-with-images solutions, as shown in Fig. 4. More visualization examples are presented in Appendix. These situations further verify the necessity of DiscPRM for scaling thinking with images.

**Comparison with Majority Voting (Self-Consistency).** We further compare DiscPRM with Majority Voting, also known as Self-Consistency, under the same candidate budget. For each test instance, we sample 8 candidate thinking-with-images trajectories from the policy model. Self-Consistency selects the final answer by majority voting over the sampled responses, while DiscPRM selects the response with the highest process reward score. This comparison allows us to examine whether the gains of DiscPRM come merely from sampling multiple candidates, or from

*Table 6.* Comparison with Majority Voting (Self-Consistency) under the same setting. DiscPRM consistently outperforms Self-Consistency across different policy models and benchmarks.

| Model | V* | MathVision | BLINK |
|---|---|---|---|
| Deepeyes | 90.1 | 26.6 | 56.3 |
| Self-Consistency | 90.9 | 27.3 | 57.1 |
| DiscPRM | **95.0** | **36.4** | **67.7** |
| Thyme-7B | 82.2 | 27.6 | 56.1 |
| Self-Consistency | 82.8 | 28.1 | 57.6 |
| DiscPRM | **89.5** | **37.1** | **68.3** |

its ability to evaluate intermediate visual-textual reasoning steps.

As shown in Tab. 6, Self-Consistency brings only marginal improvements over the original policy models. For example, with Deepeyes as the policy model, Self-Consistency improves the performance from 90.1 to 90.9 on V*, from 26.6 to 27.3 on MathVision, and from 56.3 to 57.1 on BLINK. A similar trend can be observed for Thyme-7B, where the gains of Self-Consistency are also limited. In contrast, DiscPRM achieves substantially stronger improvements under the same Best-of-8 budget. Compared with Self-Consistency, DiscPRM improves Deepeyes by 4.1, 9.1, and 10.6 points on V*, MathVision, and BLINK, respectively, and improves Thyme-7B by 6.7, 9.0, and 10.7 points.

These results indicate that the effectiveness of DiscPRM is not simply due to sampling multiple candidate responses. While Self-Consistency relies only on final-answer agreement, DiscPRM explicitly evaluates the correctness of intermediate textual and visual reasoning steps, including whether the generated visual operations provide useful evidence for the subsequent reasoning process. Therefore, process-level supervision offers additional benefits beyond final-answer voting, especially for thinking-with-images tasks where different trajectories may arrive at visually grounded or visually misleading intermediate states.

## 5. Conclusions

In this work, we develop DiscPRM, a discriminative visual PRM for test-time scaling (TTS) in thinking-with-images. To support the training of DiscPRM, we construct VTReward-100K, a dataset comprising about 100K thinking-with-images responses and their step-level reward annotations. With Best-of-N (BoN) evaluation, the DiscPRM enhances the reasoning abilities of thinking-with-images models across different tasks, outperforming prior visual PRMs.. We further introduce VABench to evaluate step-level error detection in thinking-with-images, where existing PRMs still struggle. We hope this work can inspire more future research and contribute to the development of thinking-with-images.

## Acknowledgments

This work was supported by the Fund of the National Natural Science Foundation of China under 62522607, 62495061, and 625B2093, and the Fundamental Research Funds for the Central Universities (Nankai University). This work was also supported by Alibaba Group through Alibaba Research Intern Program.

## Impact Statement

This paper presents a discriminative visual Process Reward Model (DiscPRM) for the emerging thinking-with-images paradigm, contributing to multimodal machine learning and reliable visual–textual reasoning. By jointly evaluating textual steps and intermediate visual steps and supported by the VTReward-100K supervision dataset and VABench error-detection benchmark, our framework helps reduce reasoning failures caused by incorrect intermediate visual steps and makes test-time selection strategies (e.g., Best-of-N) more effective. This can lower the technical barrier to building more dependable multimodal assistants for perception-heavy tasks and complex decision support. However, stronger visual-step reasoning and selection may also increase downstream capabilities that could be misused—for instance, enabling more convincing manipulation or interpretation of visual content, or supporting privacy-invasive analysis when applied to sensitive imagery. We advocate responsible development and deployment, including careful dataset governance and bias auditing, privacy-preserving data handling, and—when intermediate images are generated or transformed—the integration of watermarking and provenance tracking to help deter misuse and improve accountability.

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

## A. Implementation Details.

We train DiscPRM with the data-packing strategy (Bai et al., 2025). Optimization uses AdamW (Loshchilov & Hutter, 2019) with $\beta_1$=0.9, $\beta_2$=0.999, weight decay 0.05, and an initial learning rate of 1e-5. We apply a linear warmup over the first 5% of training steps, followed by cosine learning-rate decay to a minimum of 0, and train for three epochs.

## B. More Comparison Examples.

In this section, we provide more comparison examples of DiscPRM and other PRMs in Figure 6.

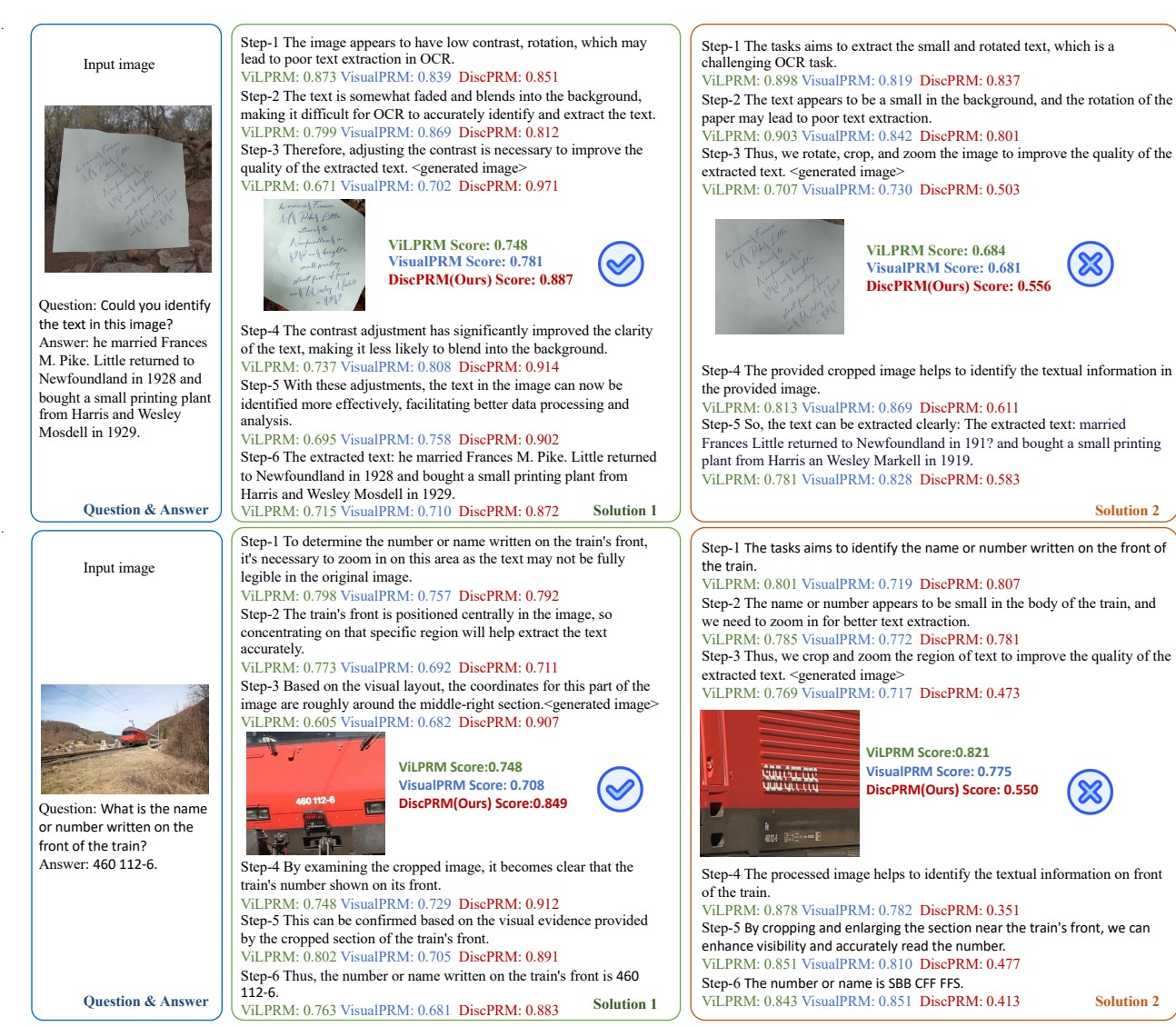

*Figure 6.* More comparison examples between our DiscPRM and other PRMs.

## C. More Data Examples in VTReward-100K.

We provide more data examples in VTReward-100K in Fig. 7.

VTReward-100K

Question: Considering the variety and presentation of the food, what might this spread suggest about the venue serving these dishes?

Answer: The refined atmosphere, paired with high-quality cuisine, creates a warm and cohesive setting—ideal for anyone seeking an upscale dining experience.

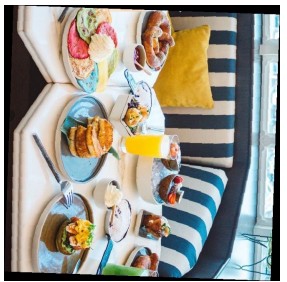

Step-1 (value=0.5625) The image appears to be rotated, making it challenging to interpret the food presentation and layout accurately.
Step-2 (value=0.6875) Rotating the image back to its original orientation will help in understanding the variety and presentation of the dishes, which is crucial for answering the user's question about the venue.
Step-3 (value=0.75) Let me proceed with the rotation to improve clarity. <generated image>

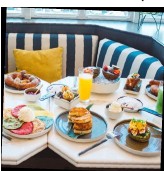

Step-4 (value=0.5) Upon examining the rotated image, the variety and presentation of the food suggest a high-end, possibly upscale venue.
Step-5 (value=0.625) The dishes are meticulously arranged, with vibrant colors and diverse textures that indicate a focus on both taste and visual appeal.
Step-6 (value=0.75) The presence of gourmet items like lobster rolls, artisanal breads, and colorful pancakes with unique toppings points to a sophisticated culinary approach.
Step-7 (value=0.625) The setting, with its elegant striped upholstery and bright, natural lighting, further reinforces the impression of a refined dining experience.
Step-8 (value=0.5625) The attention to detail in the presentation and the choice of ingredients suggest that this venue likely caters to a clientele seeking a premium dining experience.
Step-9 (value=1.0) The overall ambiance, combined with the quality of the food, creates a harmonious and inviting atmosphere, perfect for those looking to enjoy a luxurious meal.

VTReward-100K

Question: Reproduce the image text in writing.

Answer: figure fig-2 depicts histograms of the derived velocity dispersions for each the 250 kpc and 1 mpc counting radii.

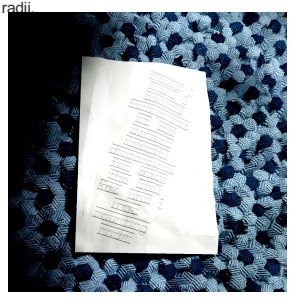

Step-1 (value=0.5625) The image appears to have low contrast, rotation, small, which may lead to poor text extraction in OCR.
Step-2 (value=0.375) The text is faint and blends into the background, making it difficult for OCR to accurately identify and extract the text.

Step-3 (value=0.3125) Enhancing the contrast, rotating and croping will make the text stand out more clearly from the background, improving readability and OCR performance.
Step-4 (value=0.0) Therefore, adjusting the image is necessary to improve the quality of the extracted text. <generated image>

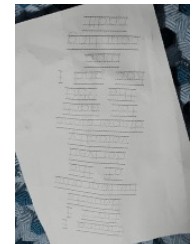

Step-5 (value=0.0) Upon analyzing the enhanced image, it is evident that the text is now more distinct and easier to read.
Step-6 (value=0.0) The contrast adjustment has significantly improved the clarity of the text, making it less likely to blend into the background.
Step-7 (value=0.0) This enhancement ensures that the text can be accurately extracted and reproduced with greater precision.
Step-8 (value=0.0) The processed image now provides a clearer visual representation, supporting more reliable OCR results.
Step-9 (value=0.0) The extracted text: histograms of the derived velocity dispersions for each of the the 7th epoch and I rms counting radius

VTReward-100K

Question: Is the bench on the left or right side of the truck?
GT Answer: Right.

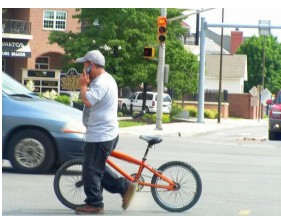

Step-1 (value=0.75) The task involves determining the position of a bench relative to a truck in the image.
Step-2 (value=0.675) The bench appears to be a small object in the scene, and its exact location is not immediately clear due to the overall composition of the image.
Step-3 (value=0.725) To accurately assess its position relative to the truck, it is necessary to zoom in and focus on the specific area where the bench is located.
Step-4 (value=0.7425) This will help in clearly identifying the bench and its spatial relationship with the truck.

Step-5 (value=0.9125) Therefore, cropping and zooming in on the region around the bench is essential for answering the question. <generated image>

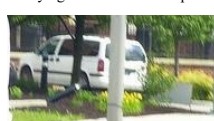

Step-6 (value=0.9875) To the left of the truck, there is a bench partially visible behind some greenery and a tree trunk.
Step-7 (value=1.00) The bench appears to be situated in a park or roadside area, and its position is clearly to the left of the truck.
Step-8 (value=1.00) Therefore, the bench is on the left side of the truck.

VTReward-100K

Question: Is the traffic light to the left of the backpack?
GT Answer: Yes.

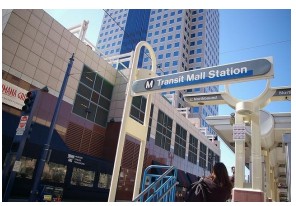

Step-1 (value=0.875) The task involves determining the relative position of the traffic light concerning the backpack.
Step-2 (value=0.6875) In the image, the traffic light is positioned on the left side.
Step-3 (value=0.725) To accurately assess their positions, it's beneficial to zoom in on the specific area around the traffic light.
Step-4 (value=0.5) This approach will provide a clearer view of their relative locations and assist in answering the question with confidence.
Step-5 (value=0.315) To achieve this, I'll crop the image focusing on the region surrounding the traffic light and the coordinates for this area seem to be near the bottom left corner of the image. <generated image>

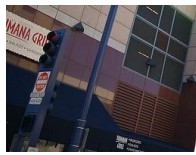

Step-6 (value=0.0) Based on the visual analysis of the provided images, I can confirm the position of the traffic light while there is no backpack.
Step-7 (value=0.0) Even in the cropped view focusing on the traffic light, there is no backpack visible.
Step-8 (value=0.0) Therefore, we cannot determine the relative positions of the traffic light and the backpack.
Step-9 (value=0.0) Final Answer: No.

*Figure 7.* More data examples from VTReward-100K.

VABench

Question: Is the cell phone on the left or right side of the bus?
GT Answer: Left.

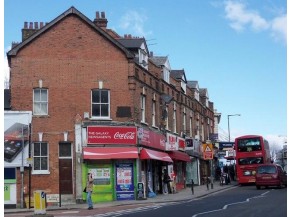

Step-1 (correct) The task involves determining if a cell phone is located on the right side of the bus.
Step-2 (correct) From the image, the bus is visible on the right side, but the details of its surroundings, including the right side, are not clearly discernible.
Step-3 (correct) To accurately assess the presence and position of the cell phone relative to the bus, it is necessary to zoom in and crop the region around the bus.
Step-4 (correct) This will allow for a clearer view of the area in question and help in making a precise determination.
Step-5 (correct) To achieve this, I'll need to crop the image around the bus and zoom in on the specific area of interest.
Step-6 (incorrect) The coordinates for this region appear to be near the right side of the image, where the bus is located.
Step-7 (incorrect) I'll refine this area further for better visibility.<generated image>

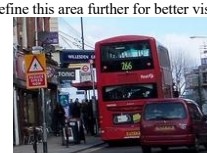

Step-8 (incorrect) Examining the cropped and zoomed-in image, it becomes clear that there is no cell phone around the bus.
Step-9 (incorrect) The area around the bus does not show any objects resembling a cell phone.
Step-10 (incorrect) Therefore, the conclusion is that there is no cell phone present on the right side of the bus.
Step-11 (incorrect) The final answer is: No, there is no cell phone.

VABench

Question: What is the cartoon character on the clock?
Answer: Mickey Mouse.

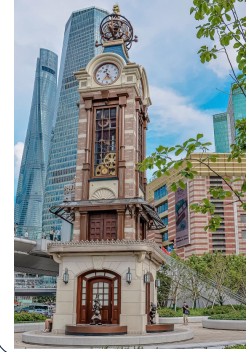

Step-1 (correct) The task involves identifying the cartoon character on the clock.
Step-2 (correct) The clock is located near the top of the clock tower in the image.
Step-3 (correct) To clearly identify the character, we need to zoom in on the clock face.
Step-4 (correct) The character appears to be a classic cartoon mouse character, which is iconic and recognizable.
Step-5 (correct) For enhanced clarity and confidence in the identification, zooming in on the clock face helps distinguish the details.
Step-6 (incorrect) The coordinates for the clock face are near the top part of the clock tower.
Step-7 (correct) I'll focus on the clock area to get a better view of the character.
Step-8 (correct) The coordinates for cropping appear to be around the center of the clock face for better detail visibility.
Step-9 (incorrect) Let me proceed with that!<generated image>

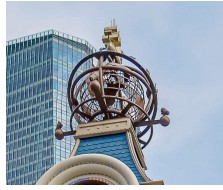

Step-10 (incorrect) The cropped and zoomed-in image of the clock face provides a clearer view of the clock, but there is no cartoon character.
Step-11 (incorrect) Thus, we cannot answer this question.

VABench

Question: In which country is this license plate registered?
Answer: Iceland.

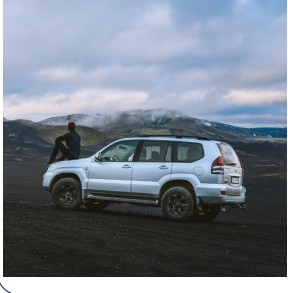

Step-1 (correct) The task is to determine the country where the license plate seen on the vehicle is registered.
Step-2 (correct) The license plate is located at the bottom left of the vehicle's rear, appearing small and somewhat unclear in the original image.
Step-3 (correct) To accurately read the text on the license plate, it is necessary to zoom in and crop this specific region.
Step-4 (correct) The coordinates for cropping are near the lower left part of the vehicle's rear section.
Step-5 (correct) By cropping and enlarging this area, we can enhance the visibility of the text on the license plate, allowing for easier identification of the country it represents. <generated image>

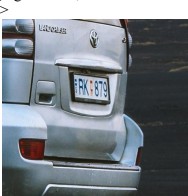

Step-6 (correct) After zooming in on the cropped image, the text on the license plate becomes legible. The visible characters \"RK 879\" correspond to the typical format used in Iceland.
Step-7 (correct) Therefore, based on this analysis, the correct option is Iceland.

VABench

Question: What is the object mounted on the wall in the background?
Answer: A notice.

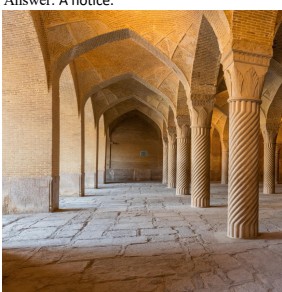

Step-1 (correct) The task is to identify the object mounted on the wall in the background.
Step-2 (correct) The image shows a large, intricately designed interior space with columns and arches.
Step-3 (correct) The object in question appears as a small, rectangular item attached to the wall near the center of the image, slightly above the midpoint. It is not immediately identifiable as one of the given options without zooming in for a closer look.
Step-4 (correct) Cropping and zooming in on this specific area will help clarify the object's details.
Step-5 (correct) The coordinates for this area seem to be around the center of the image, slightly above the middle.
Step-6 (incorrect) Cropping and zooming in on this area will help provide a clearer view of the object for better identification. <generated image>

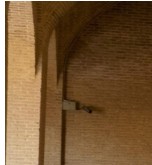

Step-7 (incorrect) The cropped and zoomed image shows that it is a light mounted on the wall in the background.
Step-8 (incorrect) Therefore, the final answer is a light.

*Figure 8.* More data examples from VABench.

# D. More Data Examples and Details for VABench.

Fig. 8 presents more data examples of VABench.

We construct VABench by curating challenging problems from five representative benchmarks: HRBench4K, VStarBench, MathVerse (Vision-Only), MME-RealWorld, and BLINK. To avoid overly easy instances and better stress-test visual-aware critics, we first filter for *challenging* samples where multiple commercial MLLMs fail to evaluate correctness of the responses as ORMs (i.e., at least one of the models produces an incorrect response selection). We then sample 200 instances from each source benchmark, forming 1,000 instances in total.

For each selected problem, we generate a step-by-step *thinking-with-images* trajectory (containing both textual and intermediate visual steps) using our policy models. Each step—textual or visual—is treated as an atomic unit to be judged for correctness, resulting in a benchmark of 1,000 responses with 9,885 total steps.

To obtain reliable step-level correctness labels at scale, we adopt a hybrid pipeline combining Monte Carlo estimation, multi-MLLM calibration, and human verification. For each step $s_i$ in a trajectory prefix $s_{\leq i}$, we first compute a Monte Carlo tree score by sampling rollouts conditioned on $s_{\leq i}$ and estimating the expected probability of reaching the correct final answer. We then convert this score into a binary label using a threshold 0.25: steps with scores above the threshold are considered correct, and otherwise incorrect.

Next, we ask three proprietary MLLMs (Qwen-VL3-Max, Gemini 3 Pro, and GPT-5.2) to independently judge the correctness of the same step. We assign a final automatic label *only when* the Monte Carlo label and all three MLLM judgments agree (all correct or all incorrect), which yields high-confidence, non-controversial annotations.

Steps with disagreement among the Monte Carlo estimate and the three MLLM judges are marked as *controversial* and are manually reviewed by three human experts with at least a university degree. Among the 9,885 total steps, 7,912 steps are non-controversial and 1,973 steps require human verification, resulting in a total workload of 8 person-days.

Overall, VABench contains 1,000 trajectories with 9,885 annotated steps, covering both visual operations and textual reasoning. We use macro-F1 over correct/incorrect steps as the primary metric to evaluate step-wise error detection.

