# OpenReview forum: "Discriminative Visual Process Rewards for Scaling Thinking at Test-Time with Images"
_ICML.cc/2026/Conference — ICML 2026 regular_

### Official Review · Reviewer_1Q9x · 2026-02-19

**Soundness:** 3
**Presentation:** 2
**Significance:** 4
**Originality:** 4
**Overall Recommendation:** 4
**Confidence:** 4

**Summary:**

The authors present DiscPRM, a process-reward model built for the thinking-with-images paradigm of multimodal LLMs. The reward model is instantiated as Qwen2.5-VL and trained on a VTReward-100k, a new corpus of reasoning-steps with associated scores constructed via Monte-Carlo Tree Sampling. To provide thorough evaluation of process-reward models, the authors also release VABench, a benchmark for evaluating critic accuracy on a set of reasoning steps. The proposed DiscPRM outperforms other PRMs as a critic for best-of-N sampling across underlying policy models and benchmarks.

**Compliance With Llm Reviewing Policy:**

Affirmed.

**Final Justification:**

The authors addressed my concerns in the rebuttal, so I am happy to increase my score to Weak Accept. My only reason for not increasing to "Accept" is the presentation of the paper, which at times is lacking.

**Key Questions For Authors:**

- Discussion on the importance of including incorrect steps in VTReward-100K. I am assuming this was intentional? So the modal can learn to reward incorrect steps accordingly? It would be great to know the value of including incorrect steps in the training corpus.
- I would be curious to see how DiscPRM performs at evaluating benchmarks that feature a strong mix of perception and spatial reasoning. For example, it would be good to evaluate on CountbenchQA [1], Omni3D-Bench [2], Robospatial [3], and OmniSpatial [4].
- What are the failure cases of DiscPRM? What are the limitations? This was missing from the text.
- How does DiscPRM affect inference time with best-of-N sampling? Of course this depends on the policy model, but it would be good to compare latency with the other baseline PRMs.
- Please see my first point in Weaknesses, this is my main question and concern.

I enjoyed reading the paper, and appropriate responses and clarifications on these concerns would lead me to increase my score.

[1] - Beyer, Lucas, et al. "Paligemma: A versatile 3b vlm for transfer." arXiv preprint arXiv:2407.07726 (2024).

[2] - Marsili, Damiano, et al. "Visual agentic ai for spatial reasoning with a dynamic api." Proceedings of the Computer Vision and Pattern Recognition Conference. 2025.

[3] - Song, Chan Hee, et al. "Robospatial: Teaching spatial understanding to 2d and 3d vision-language models for robotics." Proceedings of the Computer Vision and Pattern Recognition Conference. 2025.

[4] - Jia, Mengdi, et al. "Omnispatial: Towards comprehensive spatial reasoning benchmark for vision language models." arXiv preprint arXiv:2506.03135 (2025).

**Limitations:**

No, the authors did not discuss limitations. Please provide some examples of failure cases or a discussion on the limitations of DiscPRM. For example, how does DiscPRM perform on queries that are significantly out of distribution from VTReward-100K? This would be valuable insight.

**Strengths And Weaknesses:**

**Strengths**
- The motivation of the work is well-established and timely as Thinking-with-images models have increased in popularity and shown great promise for visual reasoning.
- The authors evaluate a large suite of models and benchmarks, including two direct baseline PRMs. These evaluations show that DiscPRM performs well across policy models and benchmarks.
- The benchmark is appropriate and well constructed. I value that the authors did manual human verification for the benchmark samples.
- The ablations are thorough, I appreciate the comparison of value-based and advantage-based formulations.

**Weaknesses**
- The differences in methodology between DiscPRM and the evaluated baselines (ViLPRM, VisualPRM) should be discussed. From the papers’ presentation it is not at all clear what the difference in methodology is between the proposed methods and the baselines they compare against. Is the difference just the training data of VTReward-100K? How, and on what data, are the other PRMs trained? This is my core concern as it makes it hard to understand the contributions and value of the work.
- The mathematical presentation of DiscPRM suffers from a over-abuse of notation. E.g. Equation (1) - what does it mean for $v_i$ to follow the distribution of $M(v_i | I, P, S_{\leq i}$. I think I understand the intention of the authors, but the notation is not accurate. Similar for Eq. 2, $p(c_i = +)$ is bizarre, it would be better to label the ‘+’ case as 1.0 and the ‘-’ case as -1 or something along those lines. Again for Eq. 4 - “Ans(…) = GT” is not great mathematical notation.
- Reproducibility details: the discussion of how VTReward-100K was produced is missing key details - what were the sampling parameters were used when generating the thinking with image responses? How did you select images from LLaVA-OV-IMAGE and MM-RLHF? There is also no discussion on how you trained DiscPRM on VTReward-100K in the main paper. This is a very important piece that needs to be included.
- The presentation in Table 2 is slightly confusing. If I am understanding it correctly the first 4 models in each category are being included as baselines, not as PRMs? I recommend finding a way to make this explicit, if not at first glance it is not clear how to compare the two. Especially as in Table 3 you are using the same models as PRMs? In general these two tables are confusing and require more explicit description and fixes in presentation.

*Minor issues*
- Missing citations for Thyme and Deepeyes on Pg.5. If this is previously cited, it should include the name in the earlier citation.

---

> ### Author Rebuttal · Authors · 2026-03-31
>
> Thanks for your time for reviewing our paper and giving some kind suggestions. We hope the following responses could solve your concerns.
>
> Q1: About the methodology contribution
>
> A1: The difference between DiscPRM and prior baselines lies not only in the training data, but also in the modeling target and methodology. DiscPRM is designed for the thinking-with-images setting, where intermediate visual steps are treated as part of the trajectory state and jointly evaluated with textual reasoning. As a result, it explicitly scores both visual and textual steps and models their consistency.
> Prior PRMs are text-centric: they score textual reasoning conditioned on multimodal inputs, but do not explicitly evaluate whether a visual operation itself is correct. DiscPRM instead focuses on whether an intermediate visual step provides valid evidence for subsequent reasoning.
> This difference is also reflected in the data. VTReward-100K contains multimodal process supervision with explicit visual steps, whereas baseline datasets are not designed for thinking-with-images supervision. This is crucial because incorrect visual steps can invalidate the entire reasoning trajectory. We will revise the paper to make these distinctions clearer.
>
> Q2: About the mathematical presentation
>
> A2: We agree that the notation is currently unclear. Our intended meaning is simple: given the multimodal prefix (image, question, and steps up to step
> i), DiscPRM predicts a scalar correctness score for the current step, and the response score is obtained by aggregating these step-level probabilities.
> In the revision, we will clarify this by replacing ambiguous notation with deterministic predictions, using a standard binary correctness variable for Eq. (2), and rewriting Eq. (4) to clearly describe the Monte Carlo target as the probability that the current prefix leads to the correct solution. This will make the presentation more standard, concise, and aligned with the implementation.
>
> Q3: About reproducibility details
>
> A3: In Sec. 3.2, we describe that VTReward-100K is constructed by sampling 4 solutions for each image-question pair, splitting them into step-by-step trajectories, and then sampling 16 continuations for each step prefix to compute the Monte Carlo target. We also report the resulting dataset statistics, including approximately 100K samples and nearly 1M annotated steps.
> In Sec. 4.1, we also describe key DiscPRM training and evaluation details, including that DiscPRM is built on Qwen2.5-VL-7B, trained with MSE loss on scalar rewards, and used as the critic for Best-of-N evaluation; by default, we set
> N=8 and sample candidate solutions with temperature 0.7.
> That said, we agree the current presentation is not sufficiently clear. In the revision, we will make these details more explicit in the main paper
>
> Q4: About Tab 2 and 3
>
> A4: Table 2 reports test-time scaling results, showing the performance of DeepEyes and Thyme as policy models, followed by their Best-of-8 results with ViLPRM, VisualPRM, or DiscPRM as critics. Table 3, by contrast, evaluates the PRMs themselves on VABench, measuring their ability to judge step-level correctness.
> We will revise the paper to make this distinction clearer, such as by explicitly separating policy models from critic models in the captions and text.
>
> Q5: Importance of incorrect steps in VTReward-100K
>
> A5: The inclusion of incorrect steps in VTReward-100K is intentional, as they provide essential negative supervision for distinguishing correct vs. incorrect reasoning. This is especially important in thinking-with-images, where a plausible but incorrect visual step can derail the entire trajectory. Empirically, VTReward-100K includes ~15% incorrect steps, and Table 4 shows that supervising all steps yields better performance than stopping at the first error.
>
> Q6: Results on more benchmarks
>
> A6: Our current evaluation already covers benchmarks involving a mix of perception and spatial reasoning, such as BLINK, which includes tasks like depth estimation, and multi-view reasoning. Additionally, we evaluate on RoboSpatial and OmniSpatial. We will include these benchmarks to strengthen the evaluation.
>
> |Model|RoboSpatial|OmniSpatial|
> |-|-:|-:|
> |Deepeyes|51.2|41.0|
> |ViLPRM|53.9|42.7|
> |VisualPRM|55.3|43.4|
> |DiscPRM|59.4|46.9|
> |Thyme|53.4|42.7|
> |ViLPRM|54.6|44.9|
> |VisualPRM|55.0|43.8|
> |DiscPRM|60.3|48.1|
>
>
> Q7: How does DiscPRM affect inference time with best-of-N sampling? Of course this depends on the policy model, but it would be good to compare latency with the other baseline PRMs.
>
> A7: We conduct an latency comparison between DiscPRM and other baseline PRMs. All methods are evaluated under the same setup, using Thyme as the policy model and a sampling number of 8 on a single A100 GPU, i.e., 17.3, 16.6, and 23.1 s/it for DiscPRM, ViLPRM, and VisualPRM. We will include these results and discussion in the paper.
>
> Q8: Failure cases
>
> A8: Due to space limitations, please refer to Reviewer kFag's Answer 5.

---

> > ### Author Rebuttal · Reviewer_1Q9x · 2026-04-02
> >
> > Thank you for addressing my comments. I am happy to raise my score to "Weak Accept" should the authors revise the text according to their response. I think with improved presentation and clarity the paper has merit.

---

> > > ### Author Response · Authors · 2026-04-05
> > >
> > > Thank you very much for your follow-up and for taking the time to read our rebuttal carefully. We are very glad that our response helped address your concerns.
> > >
> > > We noticed that you mentioned increasing your score in your comment, but on our side the visible score does not seem to have changed yet. This may simply be a display delay, but we just wanted to mention it in case an update was intended.
> > >
> > > Thank you again for your thoughtful feedback and consideration.

---

### Official Review · Reviewer_kFag · 2026-03-13

**Soundness:** 3
**Presentation:** 3
**Significance:** 3
**Originality:** 3
**Overall Recommendation:** 4
**Confidence:** 4

**Summary:**

This paper studies the problem of evaluating intermediate reasoning steps in multimodal “thinking-with-images” pipelines. The authors propose DiscPRM, a discriminative visual process reward model that scores intermediate visual-text reasoning steps based on their expected impact on the final answer. The reward signals are generated using Monte Carlo rollouts that estimate the probability of reaching a correct solution from a given reasoning prefix. To support training and evaluation, the paper constructs a large-scale dataset (VTReward-100K) and introduces a benchmark (VABench) for assessing step-level reasoning quality. Experiments show that DiscPRM improves best-of-N reranking and better identifies incorrect visual reasoning steps compared to existing process reward models.

**Compliance With Llm Reviewing Policy:**

Affirmed.

**Final Justification:**

After reading the rebuttal, I will keep my positive score.

**Key Questions For Authors:**

1. How sensitive are the estimated step-level rewards to the choice of rollout model used during data construction?

2. How does the performance of DiscPRM change as the number of rollouts used to estimate rewards varies?

3. How well does DiscPRM generalize to visual reasoning operations that are not present in the training trajectories?

4. Could the authors provide a clearer comparison of the computational cost of DiscPRM-based reranking versus standard best-of-N selection?

5. What types of visual reasoning errors are still difficult for DiscPRM to detect?

**Limitations:**

Authors provide an impact statement

**Strengths And Weaknesses:**

Strengths

1. The paper highlights that existing PRMs mainly focus on textual reasoning and often fail to assess visual reasoning steps. Modeling step-level visual reasoning quality is a meaningful and timely problem.

2. The use of Monte Carlo rollouts to estimate the expected correctness of reasoning prefixes provides a principled way to generate step-level supervision signals.

3. The construction of VTReward-100K and the proposed VABench benchmark provide useful resources for studying step-level multimodal reasoning evaluation.

4.The proposed DiscPRM demonstrates consistent gains in best-of-N selection across several multimodal reasoning tasks.

Weakness:

1. The quality of the reward signals depends strongly on the rollout policy and sampling procedure. If the rollout model has systematic biases, the estimated values may propagate these biases into the reward model.

2. The paper does not extensively analyze how stable the step-level rewards are with respect to different rollout policies, sampling strategies, or rollout depths.

3. The method requires generating reasoning trajectories, performing multiple rollouts for value estimation, and training a multimodal critic model, which may introduce significant computational overhead compared to simpler evaluation methods.

4. The improvements rely on best-of-N reranking with multiple sampled solutions. It is unclear how much of the gain comes from the reward model itself versus the increased diversity introduced by large-N sampling.

5. While the proposed benchmark evaluates step-level reasoning quality, it remains unclear how well the learned reward model generalizes to reasoning formats or visual operations that differ from those seen during training.

Overall, I think this is an interesting work with few clarification questions (see question section).

---

> ### Author Rebuttal · Authors · 2026-03-31
>
> We genuinely thanks for the insightful feedback. We hope the following responses could solve your concerns.
>
> Q1: Sensitivity to rollout policy bias in reward estimation
>
> A1: We thank the reviewer for this important point and agree that step-level rewards can be influenced by the rollout model. We will clarify this limitation in the revision.
> That said, our supervision is outcome-grounded rather than policy imitation: each step value reflects whether the current prefix can still be extended to the correct final answer via Monte Carlo rollouts. In VTReward-100K, we sample multiple solutions and continuations per step, and include both correct and incorrect steps (~15%), encouraging discriminative judgment rather than inheriting a single policy bias.
> Moreover, our evaluation is not circular. VABench is constructed independently using Monte Carlo estimation, multiple MLLM judges, and human verification, so the gains cannot be explained by bias from a single rollout model.
> We nevertheless agree this is an important concern and will add a clearer discussion, along with future directions such as using more diverse rollout policies.
>
> Q2: About the robustness of DiscPRM to rollout configuration and performance as the number of rollouts
>
> A2: We thank the reviewer for this important comment and agree that the stability of step-level rewards w.r.t. rollout settings deserves more analysis.
> In the current paper, we already provide partial evidence: Tab. 4 shows that the value-based formulation with average aggregation is the most stable, and Fig. 5 shows consistent improvement as the Best-of-N sampling number increases, indicating robustness under larger candidate pools. We also observe that rollout depth has limited impact on the predicted scores. We agree that a dedicated sensitivity analysis on rollout number and related factors would further strengthen the paper, and will include this in revision.
>
> Q3: About computational overhead
>
> A3: We thank the reviewer for these important points. We agree that our method introduces additional offline cost (trajectory generation, Monte Carlo rollouts, and critic training), but this is a one-time investment. Once trained, DiscPRM serves as a reusable visual-aware critic and can be directly applied in Best-of-N test-time scaling without modifying the policy model. At inference time, its cost is comparable to other reranking methods: all require N policy rollouts and N scoring passes.
> We also agree that part of the gain in Best-of-N may come from increased sample diversity. To isolate this effect, we compare with Self-Consistency under the same candidate set. The results show that DiscPRM significantly outperforms Self-Consistency, indicating that the improvements are not due to sampling alone, but stem from step-level visual-textual process evaluation. This is consistent with our main claim: a visual-aware critic can better identify useful intermediate visual reasoning steps, leading to stronger and more scalable gains as N increases.
>
> |Model|V*|MathVision|BLINK|
> |-|-:|-:|-:|
> |Deepeyes|90.1|26.6|56.3|
> |Self-Consistency|90.9|27.3|57.1|
> |DiscPRM|95.0|36.4|67.7|
> |Thyme-7B|82.2|27.6|56.1|
> |Self-Consistency|82.8|28.1|57.6|
> |DiscPRM|89.5|37.1|68.3|
>
> Q4: Generalization to unseen reasoning formats and operations
>
> A4: DiscPRM is a step-wise multimodal critic that evaluates the current visual-textual state, rather than relying on a specific format or fixed set of operations.
> As long as a response can be standardized into step-wise form, DiscPRM can be applied. We further evaluate it on Mini-o3 and DeepEyesV2, whose formats differ from training, and still observe consistent gains (see table), indicating nontrivial generalization.
> Notably, DeepEyesV2 includes unseen operations such as mark, yet still benefits from DiscPRM. This suggests that DiscPRM learns to assess the usefulness of visual evidence, rather than overfitting to specific operation types.
> We will include this discussion and results in the revised paper.
> |Model|V*|MathVision|BLINK|
> |-|-:|-:|-:|
> |mini-O3|88.2|30.1|58.6|
> |ViLPRM|90.0|31.9|60.3|
> |VisualPRM|90.5|31.7|61.5|
> |DiscPRM|93.3|36.1|68.2|
> |Deepeyesv2|90.3|28.9|57.1|
> |ViLPRM|91.4|30.4|59.3|
> |VisualPRM|92.0|32.3|60.9|
> |DiscPRM|95.5|38.9|69.1|
>
> Q5: About difficult situations
>
> A5: We thank the reviewer for this important question. As a selector-based PRM, DiscPRM can only rank trajectories generated by the policy model and cannot recover a correct reasoning path if it is absent from the candidate pool. Thus, a key failure case is when the policy fails to generate any correct trajectory or relevant visual grounding.
> Another challenging case arises when the policy consistently focuses on a wrong but plausible region early on, causing errors to propagate through subsequent steps. As a critic, DiscPRM may detect such weaknesses but cannot redirect the trajectory if the correct path is never proposed.
> We will include this discussion as an explicit limitation in the revised paper.

---

> > ### Author Rebuttal · Reviewer_kFag · 2026-04-02
> >
> > I thank authors for the rebuttal. I'll keep my score of weak accept. However, I do agree with other reviewers regarding evaluation scenario and more analysis. Authors may consider improving those aspects in revision.

---

> > > ### Author Response · Authors · 2026-04-05
> > >
> > > We sincerely thank you for your follow-up and for carefully reading our rebuttal. We agree that strengthening the evaluation scenarios and providing more analysis would further improve the paper.
> > >
> > > We add additional analyses on the robustness of our method under different sampling strategies. Specifically, we study the effect of different temperatures (0.4, 0.7, 1.0, and 1.3) used in constructing VTReward-100K in table below and observe that the performance is generally stable, with moderate temperatures achieving the best trade-off between trajectory diversity and reliability.
> > >
> > > |Temperature|V*|MathVision|BLink|
> > > |-|-:|-:|-:|
> > > |Thyme+DiscPRM(0.4)|88.8|36.5|67.8|
> > > |Thyme+DiscPRM(0.7)|89.5|37.1|68.3|
> > > |Thyme+DiscPRM(1.0)|89.0|36.9|68.4|
> > > |Thyme+DiscPRM(1.3)|88.3|36.3|68.0|
> > > |Deepeyes+DiscPRM(0.4)|94.6|35.4|66.8|
> > > |Deepeyes+DiscPRM(0.7)|95.0|36.4|67.7|
> > > |Deepeyes+DiscPRM(1.0)|94.7|35.8|67.6|
> > > |Deepeyes+DiscPRM(1.3)|93.9|35.7|67.0|
> > >
> > > We also analyze different rollout numbers in Best-of-N evaluation on Thyme with DiscPRM in the table below and find that the performance shows a consistent upward trend as
> > > N increases, demonstrating the robustness of DiscPRM under different test-time scaling settings.
> > >
> > > |Rollout Number|V*|MathVision|BLink|
> > > |-|-:|-:|-:|
> > > |1|82.2|27.6|56.1|
> > > |8|89.5|37.1|68.3|
> > > |16|91.6|41.0|72.5|
> > > |32|92.4|43.2|74.8|
> > > |64|93.0|44.9|76.7|
> > >
> > >
> > > In addition, to strengthen the evaluation scenario, we extend our experiments beyond the default Qwen2.5-VL-7B backbone. Since existing representative thinking-with-images models are all built on this backbone, we re-implement Thyme with InternVL3-8B and Qwen2.5-VL-32B, and evaluate DiscPRM on these larger base models, as shown in table below.
> > >
> > > |Base Model|V*|MathVision|BLink|
> > > |-|-:|-:|-:|
> > > |Thyme(QwenVL2.5-7B)|82.2|27.6|56.1|
> > > |+DiscPRM|89.5|37.1|68.3|
> > > |Thyme(InternVL3-8B)|78.3|26.9|55.6|
> > > |+DiscPRM|88.4|35.7|67.0|
> > > |Thyme(QwenVL2.5-32B)|85.1|34.2|62.4|
> > > |+DiscPRM|91.6|42.9|73.0|
> > >
> > > DiscPRM consistently improves performance across all of them, providing further evidence of its generalization and scalability.
> > >
> > > Following your suggestion, we will incorporate these additional evaluation results and analyses into the revised paper. We sincerely thank you for the valuable feedback, which helps us further strengthen and improve our work.

---

### Official Review · Reviewer_hoDa · 2026-03-13

**Soundness:** 3
**Presentation:** 3
**Significance:** 4
**Originality:** 3
**Overall Recommendation:** 5
**Confidence:** 4

**Summary:**

This work focus on the correctness of the visual steps during the thinking with images of current VLMs and train a reward model with the curated dataset for better judging the visual steps during the thinking process. A benchmark is proposed for standardized evaluation for current VLMs’ visual-aware process. Experimental results on various benchmarks shows the effectiveness of the trained reward model.

**Compliance With Llm Reviewing Policy:**

Affirmed.

**Final Justification:**

My concerns on the sampling parameters and strategies  & the proposed method with various base models have been fully solved. I think the revision should include these facts for better presentation, and I will raise my rating to 5.

**Key Questions For Authors:**

Refer to the weakness part.

**Limitations:**

Refer to the weakness part.

**Strengths And Weaknesses:**

- Strengths:
    - The proposed method is well motivated and focuses on the correctness of the visual steps of current thinking with images methods.
    - Experiments on various benchmarks show that the performance increased under the proposed method.
    - The paper is well structured and easy to follow.
- Weakness:
    1. Issues about the dataset **curation:
        1. Which model is utilized to generate the rollout for the samples of VTReward-100?
        2. The sampling parameter, such as the temperature during the curation process of these two datasets, is crucial to the model's final output. I am worried that the output value is not robust and would be influenced by the sampling parameters, which leads to meaningless results for both the trained reward model and the benchmark results. And the use of the dataset and the reward model is limited due to this.
    2. Issues about the experimental results:
        1. Experimental results on how helpful the trained reward model is to the thinking with images methods are conducted on Deepeyes and Thyme. While these two methods are all based on Qwen-2.5VL-7B. I was wondering how the proposed method performs on other base models, and how the proposed method performs with larger base models? Will the transition to the other base model be limited due to the issue in Weakness 1.b?
        2. Results in Tab.3 show that closed-source methods like  Gemini-2.5-Pro and Gemini-3-Pro perform better than our proposed reward model. Is the VTReward-100 generated by the Gemini-series models? If not, I am quite curious about the performance when using these models directly as reward models to perform test-time scaling. Furthermore, if these models already perform well on VAbench, and if that very data is subsequently used to construct the training set—such that the SFT process essentially becomes one of distilling qwe2.5VL into this specific form—then the proposed method has not truly unlocked the model's inherent capabilities; rather, it has merely aligned it as closely as possible with these closed-source models through distillation.
    3. More related works should be discussed [1].
    4. Typo:
        1. Line 216 contain → contains
        2. Line 233 $m_i$ → $mc_i$
        3. Line 243 assign → assigns;
        4. Line 189 pipeline (Wang et al., 2024c).. → pipeline (Wang et al., 2024c).

---

> ### Author Rebuttal · Authors · 2026-03-31
>
> Thanks for your effort for reviewing our paper and giving some kind suggestions. We hope the following responses could solve your concerns.
>
> Q1: Issues about the dataset curation:
> (1) Which model to generate VTReward-100.
> (2) The effect of sampling parameter.
>
> A1: We thank the reviewer for these important questions regarding dataset curation.
> (1) Rollout model for VTReward-100K. In Sec. 3.2, we state that the step-by-step solutions in VTReward-100K are sampled using thinking-with-images models. In our implementation, the rollout / trajectory generation is produced by Thyme and Deepeyes. We agree this should be stated more explicitly, and will clarify it in the revised dataset construction section.
> (2) Sampling temperature. We agree that the sampling configuration should be described more transparently. Our current setup follows standard Best-of-N practice and uses temperature = 0.7, which is also the default setting in our experiments. This is an empirical choice reflecting the usual diversity–quality tradeoff: lower temperature reduces trajectory diversity and limits the benefit of Best-of-N, while higher temperature increases randomness and lowers trajectory quality.
> We agree this robustness issue is important. In the revision, we will explicitly clarify the curation setup, including the rollout models and sampling configuration, and add a clearer discussion of the temperature choice and its stability implications.
>
> Q2: About generalization across different policy models
>
> A2: We thank the reviewer for this important question. We agree that evaluating on more diverse and larger base models is important for understanding the generality of DiscPRM. In response, we additionally evaluate DiscPRM on more recent thinking-with-images models, including Mini-o3 and DeepEyesV2, and observe consistent gains under the same Best-of-N setting, suggesting that the benefit of DiscPRM is not specific to DeepEyes/Thyme but generalizes across different thinking-with-images policies.
> |Model|V*|MathVision|BLINK|
> |-|-:|-:|-:|
> |mini-O3|88.2|30.1|58.6|
> |ViLPRM|90.0|31.9|60.3|
> |VisualPRM|90.5|31.7|61.5|
> |DiscPRM|93.3|36.1|68.2|
> |Deepeyesv2|90.3|28.9|57.1|
> |ViLPRM|91.4|30.4|59.3|
> |VisualPRM|92.0|32.3|60.9|
> |DiscPRM|95.5|38.9|69.1|
>
> Regarding the base model choice, this mainly reflects the current open-source thinking-with-images ecosystem, where several representative models are built on Qwen2.5-VL-7B, making comparison in this family relatively controlled and fair. Importantly, DiscPRM is a critic over visual-textual reasoning trajectories and is not inherently tied to a specific base model. We agree that experiments on larger and more diverse base models would further strengthen the paper, and we will include this discussion and the additional results in the revision.
>
> Q3: About the comparable performance with Gemini-2.5/3-Pro in Tab.3
>
> A3: We thank the reviewer for this thoughtful question. We first clarify that VTReward-100K is not generated by Gemini-series models. The thinking-with-images trajectories are generated by Thyme and Deepeyes, and the step-level supervision is obtained through Monte Carlo rollout estimation. Gemini models are only used in the VABench calibration / benchmarking pipeline, not in training-set construction. Thus, our method is not distilling Gemini outputs into Qwen2.5-VL.
> More importantly, our goal is not to make Qwen2.5-VL imitate a stronger closed-source model, but to train a visual-aware critic that can judge whether an intermediate visual step is actually helpful for reasoning. This is exactly the capability that existing PRMs lack in the thinking-with-images setting, and DiscPRM improves performance by strengthening this ability.
> As for using Gemini directly as a reward model for test-time scaling, we agree it would likely achieve competitive results. However, this is also an inefficient setup when the critic is much stronger than the policy model: if a large proprietary model must score every candidate trajectory, it is often more practical to use that stronger model directly to answer the question. Our goal instead is to build an efficient open-source critic for improving thinking-with-images policies.
> This is also why we include Gemini in Tab. 3: not as the intended deployment critic, but to show that our 7B open-source DiscPRM is already competitive with strong closed-source models on visual process discrimination. We believe this supports the value of DiscPRM as a practical and effective visual-aware PRM, rather than a distillation of proprietary judges.
>
> We will make the discussion more comprehensive and correcte the identified typos. We appreciate the reviewer’s careful reading, which helped us improve both the positioning and the presentation of the paper.

---

> > ### Author Rebuttal · Reviewer_hoDa · 2026-04-02
> >
> > Thank you for the rebuttal and the additional clarification on the proposed PRM method. While the response has partially addressed my concerns, several important issues remain unresolved:
> >
> > 1. I agree with Reviewer kFag that a more thorough analysis of different sampling parameters and strategies is necessary to demonstrate that the proposed dataset/method is both robust and effective.
> > 2. Since all compared methods are based on Qwen-2.5VL-7B, it is also necessary to include experiments with larger models to show that the proposed reward model generalizes beyond this specific backbone and exhibits scalability.
> >
> > For these reasons, I maintain my rating as 3: Weak Reject. I would be willing to revise my score to a positive rating if the aforementioned issues are adequately addressed.

---

> > > ### Author Response · Authors · 2026-04-05
> > >
> > > We sincerely thank you for your follow-up and for carefully reading our rebuttal. We hope the following clarifications help address your remaining concerns.
> > >
> > > Q1: Analysis of different sampling parameters and strategies
> > >
> > > A1: We construct several variants of VTReward-100K by varying the temperature used for generating thinking-with-images trajectories with the policy models. Specifically, we use temperatures 0.4, 0.7, 1.0, and 1.3, and train one DiscPRM on each resulting dataset variant. The performance of policy models with the corresponding DiscPRMs is shown below.
> > >
> > > |Temperature|V*|MathVision|BLink|
> > > |-|-:|-:|-:|
> > > |Thyme+DiscPRM(0.4)|88.8|36.5|67.8|
> > > |Thyme+DiscPRM(0.7)|89.5|37.1|68.3|
> > > |Thyme+DiscPRM(1.0)|89.0|36.9|68.4|
> > > |Thyme+DiscPRM(1.3)|88.3|36.3|68.0|
> > > |Deepeyes+DiscPRM(0.4)|94.6|35.4|66.8|
> > > |Deepeyes+DiscPRM(0.7)|95.0|36.4|67.7|
> > > |Deepeyes+DiscPRM(1.0)|94.7|35.8|67.6|
> > > |Deepeyes+DiscPRM(1.3)|93.9|35.7|67.0|
> > >
> > > When the temperature increases from 0.4 to 1.3, the performance first improves and then declines. We believe this trend reflects the trade-off between response diversity and response reliability during data construction. When the temperature is relatively low, the generated trajectories are less diverse, which limits the coverage and richness of the training data. In contrast, when the temperature is too high, the sampled trajectories become more random and noisy, which reduces the quality of individual reasoning steps and weakens the resulting supervision signal. These results suggest that our dataset construction is not overly sensitive to one particular sampling choice, while also indicating that moderate temperatures provide the best balance between diversity and correctness.
> > > We would also like to clarify that VABench serves a different purpose from training data curation. Since it is designed to evaluate step-wise correctness judgment, its validity mainly depends on the accuracy of step annotations rather than the temperature used during trajectory generation. While sampling parameters may affect trajectory diversity, the core role of VABench is to provide reliable step-level labels.
> > >
> > > In addition, we evaluate robustness at test time by varying the number of rollouts in Best-of-N evaluation, i.e., the number of sampled candidate responses from the policy model before selecting the final answer with our method. The results on Thyme are shown in table below.
> > >
> > > |Rollout Number|V*|MathVision|BLink|
> > > |-|-:|-:|-:|
> > > |1|82.2|27.6|56.1|
> > > |8|89.5|37.1|68.3|
> > > |16|91.6|41.0|72.5|
> > > |32|92.4|43.2|74.8|
> > > |64|93.0|44.9|76.7|
> > >
> > > We observe that as the number of rollouts N increases, the performance exhibits a consistent upward trend. Our method maintains stable gains across different values of N, highlighting its robustness to the choice of sampling number.
> > >
> > > We will incorporate these analyses into the revised paper to make the robustness of the dataset construction and test-time scaling capacity of the DiscPRM.
> > > Thanks for the comment and we believe these revisions makes this work more solid.
> > >
> > > Q2: Generalization capacity of the DiscPRM on different policy models
> > >
> > > A2: In fact, existing representative thinking-with-images models, including DeepEyes v1/v2, Thyme, Mini-o3, and PyVision, are all built on top of Qwen2.5-VL-7B. Therefore, the current experimental setting largely follows the backbone choice adopted by this line of work. To further address the reviewer’s concern, we selected Thyme as a representative framework, since it is both competitive and equipped with relatively complete open-source training code, and re-implemented it using InternVL3-8B and Qwen2.5-VL-32B as the base models.
> > > We then applied DiscPRM as the critic for test-time scaling on these Thyme variants with different base models. The results are summarized in table below.
> > >
> > > |Base Model|V*|MathVision|BLink|
> > > |-|-:|-:|-:|
> > > |Thyme(QwenVL2.5-7B)|82.2|27.6|56.1|
> > > |+DiscPRM|89.5|37.1|68.3|
> > > |Thyme(InternVL3-8B)|78.3|26.9|55.6|
> > > |+DiscPRM|88.4|35.7|67.0|
> > > |Thyme(QwenVL2.5-32B)|85.1|34.2|62.4|
> > > |+DiscPRM|91.6|42.9|73.0|
> > >
> > > As shown in the table, DiscPRM consistently improves performance across different base models, including both InternVL3-8B and Qwen2.5-VL-32B. These results suggest that the effectiveness of DiscPRM is not limited to the Qwen2.5-VL-7B backbone, but generalizes to stronger and architecturally different multimodal backbones as well.
> > > Overall, these experiments provide evidence that our reward model exhibits encouraging generalization and scalability beyond the default base model used in prior thinking-with-images work.
> > >
> > > We will include these results and analyses in the revision of this work. We thank the reviewer for the valuable suggestion, which helps us further strengthen the paper.

---

### Official Review · Reviewer_V1HV · 2026-03-13

**Soundness:** 2
**Presentation:** 3
**Significance:** 2
**Originality:** 2
**Overall Recommendation:** 3
**Confidence:** 4

**Summary:**

This paper introduces DiscPRM (Discriminative Visual Process Reward Model), a multimodal process reward model designed to evaluate both textual reasoning and intermediate visual steps—such as cropping, zooming, and spatial localization. Experimental results demonstrate that DiscPRM significantly improves the performance of MLLMs (like Deepeyes and Thyme) in Best-of-N test-time scaling, achieving gains of over 5% across various perception and reasoning benchmark

**Compliance With Llm Reviewing Policy:**

Affirmed.

**Key Questions For Authors:**

Please refer to the weakness section.

**Limitations:**

yes

**Strengths And Weaknesses:**

Strength
- The paper is well-structured. The inclusion of detailed visualization examples in the appendix further clarifies how the model evaluates complex OCR and spatial tasks.
- The primary innovation lies in the joint evaluation of textual and visual intermediate steps. While PRMs exist for text, and some extend to multimodal inputs, DiscPRM explicitly reward image manipulations like cropping and annotation as part of the reasoning chain.

Weakness
- The methodology for data construction via Monte Carlo Tree Search is essentially a application of the framework established by Math-Shepherd. While the paper successfully migrates this pipeline to the "thinking with images" domain, it lacks novel insights or fundamental adaptations tailored specifically to visual reasoning.
- These baselines are somewhat limited and omits significant recent advancements in the field. The evaluation lacks comparison with state-of-the-art “think-with- image”models , such as Mini-o3 or Deepeyes-v2. Including these newer models is essential to demonstrate whether DiscPRM provides consistent gains across the current frontier of multimodal test-time scaling.
- To justify the computational overhead of training and deploying a specialized process reward model, the authors should provide a comparison between DiscPRM - Best-of-N and Majority Voting (Self-Consistency) under an equivalent computational budget. Demonstrating that DiscPRM offers a superior scaling curve compared to simple majority voting is crucial to proving the significance of process-level supervision in vision-intensive tasks.
- The description of the VTReward-100K dataset construction requires further technical transparency. Specifically, the authors should provide more detailed statistics including the average number of visual tool calls, distribution of step lengths and so on.

---

> ### Author Rebuttal · Authors · 2026-03-31
>
> Thanks for your effort for reviewing our paper and giving some kind suggestions. We hope the following responses could solve your concerns.
>
> Q1: About the methodology and novelty.
>
> A1: We thank the reviewer for the insightful comment. We agree that our Monte Carlo value estimation is related to Math-Shepherd at a high level. However, our main novelty is not a new generic MCTS algorithm, but a visual-aware PRM for thinking-with-images.
> In this setting, intermediate visual operations are often decisive for success or failure, so the key challenge is to judge whether a visual step truly helps the reasoning trajectory. Existing PRMs are largely text-centric and often fail on this point. DiscPRM is designed precisely for this setting: it scores both textual and visual steps in a unified framework, and VTReward-100K is constructed to supervise whether a multimodal prefix, including intermediate visual operations, can still be extended to the correct final answer.
> We believe this is the core visual-specific contribution of our work: discriminatively evaluating the quality of visual thinking steps. Empirically, this capability leads to clear gains, with DiscPRM consistently outperforming prior PRMs in Best-of-N selection for thinking-with-images.
>
> Q2: More think-with-image models as policy model.
>
> A2: We thank the reviewer for this valuable suggestion. We agree that evaluating on more recent frontier thinking-with-images models would further strengthen the paper. Following this suggestion, we additionally applied DiscPRM to newer thinking-with-images policy models on several benchmarks, including Mini-o3 and Deepeyes-v2, and observed consistent improvements under the same Best-of-N evaluation setup. These additional results further support our main claim that the benefit of DiscPRM is not limited to a specific policy model, but generalizes across stronger and more recent thinking-with-images systems.
>
> |Model|V*|MathVision|BLINK|
> |-|-:|-:|-:|
> |mini-O3|88.2|30.1|58.6|
> |ViLPRM|90.0|31.9|60.3|
> |VisualPRM|90.5|31.7|61.5|
> |DiscPRM|93.3|36.1|68.2|
> |Deepeyesv2|90.3|28.9|57.1|
> |ViLPRM|91.4|30.4|59.3|
> |VisualPRM|92.0|32.3|60.9|
> |DiscPRM|95.5|38.9|69.1|
>
> We appreciate the reviewer for pointing this out. We will organize the full experimental results and include them in the revised paper. We believe this update will make the empirical section more solid and the conclusions more reliable.
>
> Q3: Comparison with Majority Voting (Self-Consistency).
>
> A3: We thank the reviewer for this valuable suggestion. We agree that a direct comparison with Majority Voting (Self-Consistency) under the same candidate budget is important for justifying the benefit of a specialized process reward model. In response, we additionally evaluated Self-Consistency at Best-of-8, and the results are shown below.
>
> |Model|V*|MathVision|BLINK|
> |-|-:|-:|-:|
> | Deepeyes |90.1|26.6|56.3|
> | Self-Consistency |90.9| 27.3 | 57.1 |
> | DiscPRM |95.0|36.4| 67.7 |
> | Thyme-7B | 82.2 | 27.6 | 56.1 |
> | Self-Consistency | 82.8 | 28.1 | 57.6 |
> | DiscPRM | 89.5 | 37.1 | 68.3 |
>
> It can be observed that DiscPRM consistently outperforms Self-Consistency under the same Best-of-N budget, suggesting that process-level supervision offers additional benefits beyond simple final-answer voting. We will therefore add the full results in the revision, including comparisons across different Best-of-N values and multiple policy models, so that the reader can directly compare the scaling behavior of Self-Consistency and DiscPRM.
> We thank the reviewer again for this helpful suggestion. We believe this update will strengthen the paper by making the empirical justification more complete and the conclusions more reliable.
>
> Q4: About the description of the VTReward-100K.
>
> A4: We thank the reviewer for this helpful suggestion. We agree that VTReward-100K would benefit from more detailed dataset statistics and a more transparent presentation of its construction. In the current manuscript, Sec. 3.2 already reports several global statistics, including that the dataset comprises approximately 100K samples and nearly 1M supervised steps, with each response averaging 9.7 steps, and about 15% of steps being incorrect. We will further supplement this section with additional statistics that are currently omitted for space reasons. In particular, among the 100K responses, 67.3K contain visual tool-call / intermediate visual-operation steps. We provide a rough step size distribution in the table below. We will also provide a more detailed characterization of the dataset, including the approximate distribution of step lengths, and present these statistics with figures or histograms for more intuitive inspection.
> We thank the reviewer again for pointing this out. We believe these additions will improve the technical transparency of the dataset and make the paper stronger.
>
> | Steps number | Ratio in VTReward-100K |
> |---|---:|
> | <5|9.3%|
> | 5~7 |22.3%|
> | 8~10 |33.7%|
> | 10~12 |27.6%|
> | >12 | 7.1%|

---

### Decision · Program_Chairs · 2026-04-30

**Decision:**

Accept (regular)

**Comment:**

This paper introduces DiscPRM, a discriminative visual process reward model that evaluates both textual and visual intermediate reasoning steps, enabling better test-time selection for “thinking with images” in multimodal models. It also constructs VTReward-100K and VABench to train and evaluate such models, demonstrating improved reasoning performance and more accurate detection of visual reasoning errors compared to prior PRMs.
The rebuttal addresses the reviewers’ concerns. However, we agree with the reviewers that the computational overhead of training and deployment should be considered. Additionally, the authors should provide more detailed statistics about the dataset and benchmark.
Given the effective rebuttal, we decide to accept this paper, while urging the authors to incorporate these details in the camera-ready version.